# Tgif1-deficiency impairs cytoskeletal architecture in osteoblasts by activating PAK3 signaling

**Simona Bolamperti**[1], **Hiroaki Saito**[1,2,3], **Sarah Heerdmann**[1], **Eric Hesse**[1,2,3†], **Hanna Taipaleenmäki**[1,2,3*†]

[1]Molecular Skeletal Biology Laboratory, Department of Trauma Surgery and Orthopedics, University Medical Center Hamburg-Eppendorf, Hamburg, Germany; [2]Institute of Musculoskeletal Medicine, LMU University Hospital, LMU Munich, Munich, Germany; [3]Musculoskeletal University Center Munich, LMU University Hospital, LMU Munich, Munich, Germany

**\*For correspondence:**
hanna.taipaleenmaeki@med.uni-muenchen.de

†These authors contributed equally to this work

**Competing interest:** The authors declare that no competing interests exist.

**Abstract** Osteoblast adherence to bone surfaces is important for remodeling bone tissue. This study demonstrates that deficiency of TG-interacting factor 1 (Tgif1) in osteoblasts results in altered cell morphology, reduced adherence to collagen type I-coated surfaces, and impaired migration capacity. Tgif1 is essential for osteoblasts to adapt a regular cell morphology and to efficiently adhere and migrate on collagen type I-rich matrices in vitro. Furthermore, Tgif1 acts as a transcriptional repressor of p21-activated kinase 3 (*Pak3*), an important regulator of focal adhesion formation and osteoblast spreading. Absence of Tgif1 leads to increased *Pak3* expression, which impairs osteoblast spreading. Additionally, Tgif1 is implicated in osteoblast recruitment and activation of bone surfaces in the context of bone regeneration and in response to parathyroid hormone 1–34 (PTH 1–34) treatment in vivo in mice. These findings provide important novel insights in the regulation of the cytoskeletal architecture of osteoblasts.

## eLife assessment

This **important** work substantially advances our understanding of osteoblast migration to the sites of bone formation and regeneration. The evidence supporting the conclusion is **compelling**, with rigorous in vitro assays for cellular and biochemical aspects and with appropriate in vivo models. The work will be of broad interest to developmental biologists and bone biologists.

## Introduction

Bone remodeling is a highly coordinated and continuously ongoing process involving osteoblasts, osteoclasts, and osteocytes, that interact with each other and function in concert to maintain bone mass and preserve the integrity of the skeletal system (*Baron and Hesse, 2012*). This process is essential for bone development, growth, maintenance, and repair as well as for the functionality of pharmacological interventions (*Kenkre and Bassett, 2018*). Osteoblasts, derived from mesenchymal stem cells, play a pivotal role in synthesizing and depositing collagen type I-rich bone matrix (*Ponzetti and Rucci, 2021*). To achieve this, osteoblasts undergo changes in cell shape and migrate on bone surfaces to lay down matrix at places where new bone needs to be formed, necessitating the regulation of cell movement and shape acquisition (*Jones and Boyde, 1977*; *Eleniste et al., 2014*; *Thiel et al., 2018*).

Migration of osteoblasts involves movement from their origin to specific locations within the bone tissue and occurs during both embryonic development and adult bone remodeling. Multiple factors influence osteoblast migration, including growth factors, cytokines, signaling pathways like bone morphogenetic proteins (BMPs) and transforming growth factor-beta (TGF-β) as well as interactions with other cells and the extracellular matrix (*Thiel et al., 2018*). These signals not only promote migration but also regulate the expression of adhesion molecules, cytoskeletal rearrangements, and cellular contractility.

Osteoblasts exhibit various morphological shapes based on their differentiation stage and the local microenvironment, ranging from elongated spindle-like morphology to cuboidal appearance (*Tsuji et al., 2022*). The cytoskeleton, primarily composed of actin filaments, microtubules, and intermediate filaments, plays a central role in determining cell shape. Actin filaments, among other cytoskeletal elements, are involved in cell polarization, protrusion formation, and contractility during migration (*Schaks et al., 2019*; *Tang and Gerlach, 2017*; *Etienne-Manneville, 2004*). Signaling molecules, including Rho GTPases such as RhoA, Rac1, and Cdc42, regulate the cytoskeleton by controlling actin polymerization and organization, contributing to the formation of structures like lamellipodia, filopodia, and stress fibers (*Schaks et al., 2019*). These structures are essential for cell migration, enabling attachment to the extracellular matrix and generating the necessary force for movement. Integrins, transmembrane proteins mediating cell-extracellular matrix interactions, also play a critical role in osteoblast migration and cell shape regulation. They anchor cells to the matrix, transmit signals regulating migration and cytoskeletal dynamics, and activate intracellular pathways affecting cell shape, migration speed, and adhesion strength (*Kechagia et al., 2019*; *SenGupta et al., 2021*; *Hood and Cheresh, 2002*; *Thiel et al., 2018*). Thus, osteoblast migration and cell shape acquisition are interconnected processes crucial for bone development, growth, remodeling, and repair. Further investigation of the mechanisms underlying osteoblast migration and cell shape regulation is, therefore, important to better understand bone mass maintenance, treatment of bone-related disorders, and bone repair.

Fractures occur in the context of high-energy trauma but also due to bone fragility and represent a severe damage of bone integrity, requiring subsequent tissue regeneration. In response to fracture, inflammatory cells including neutrophils and macrophages are recruited to the site, releasing signaling molecules such as cytokines and growth factors. These molecules create a favorable environment for osteoblast migration and bone formation (*Einhorn and Gerstenfeld, 2015*). Osteoblast precursors derived from mesenchymal stem cells, or the periosteum migrate alongside sprouting vessels towards the fracture gap and differentiate into mature bone-forming osteoblasts (*Maes et al., 2010*). Once osteoblasts have reached the fracture site, production and deposition of new bone matrix is initiated to facilitate fracture repair (*Maes et al., 2010*; *Einhorn and Gerstenfeld, 2015*). During this process, chemotactic signals released by surrounding tissues and cells, including platelet-derived growth factor (PDGF), TGF-β, and BMPs guide osteoblast migration (*Dirckx et al., 2013*; *Einhorn and Gerstenfeld, 2015*; *Thiel et al., 2018*). Furthermore, the migration of osteoblasts is facilitated by the dynamic rearrangement of the cytoskeleton, primarily composed of actin filaments (*Thiel et al., 2018*). Structures like lamellipodia and filopodia, enriched with actin, enable osteoblasts to extend and protrude in the direction of migration, interacting with the extracellular matrix and providing necessary traction (*Casati et al., 2014*; *Jafari et al., 2019*).

Movement of osteoblasts is an important but often underestimated component of the pharmacologically induced gain in bone mass in response to the treatment with bone anabolic drugs. Teriparatide, a recombinant form of the first 34 amino acids of human parathyroid hormone (rhPTH1-34; hereafter PTH), is a bone anabolic drug used to treat severe osteoporosis, characterized by low bone mineral density and an increased fracture risk (*Neer et al., 2001*; *Black and Rosen, 2016*). PTH treatment stimulates bone formation by increasing the number, differentiation, and activity of osteoblasts (*Pettway et al., 2008*; *Silva et al., 2011*; *Ogita et al., 2008*). In addition, PTH promotes the recruitment of osteoblast precursor cells and activates quiescent lining cells on bone surfaces, transforming them into active matrix-forming osteoblasts (*Nishida et al., 1994*; *Dobnig and Turner, 1995*; *Kim et al., 2012*). Furthermore, PTH increases the migration of mesenchymal precursor cells (*Lv et al., 2020*) and augments bone remodeling (*Saito et al., 2019*), which attracts osteoblasts (*Dirckx et al., 2013*). During osteoblast migration, cytoskeletal dynamics promote actin filament remodeling and the formation of filopodia and lamellipodia (*Lomri and Marie, 1990*). These structures are essential for

cell attachment to the extracellular matrix during migration. By enhancing cytoskeletal dynamics, PTH facilitates osteoblast movement (*Thiel et al., 2018*). PTH also increases the expression and activation of integrins, which mediate interactions between cells and the extracellular matrix (*Gronthos et al., 2001*; *Kaiser et al., 2001*). These combined pharmacological effects of PTH treatment result in an increase in bone formation and bone mineral density, which improves bone strength and ultimately reduces fracture risk (*Neer et al., 2001*; *Black and Rosen, 2016*).

Recently, we reported the important role of the homeodomain protein TG-interacting factor 1 (Tgif1) in osteoblast differentiation, activity, and bone formation (*Saito et al., 2019*). Furthermore, *Tgif1* was identified as PTH target gene, and in the absence of Tgif1, PTH treatment failed to increase bone mass in mice (*Saito et al., 2019*). These findings demonstrate the importance of Tgif1 as a novel regulator of bone remodeling and emphasize its essential involvement in mediating the full bone anabolic effect of PTH treatment. Building upon these observations, we investigated the role of Tgif1 in osteoblast morphology, adherence, and migration. Our findings demonstrate that Tgif1-deficient osteoblasts display an altered morphology, reduced adherence to collagen type I-coated surfaces, impaired migration capacity, and decreased spreading compared to control cells. These deficits are associated with a compromised formation of focal adhesions and an upregulated expression of p21-activated kinase 3 (PAK3), which we further investigated due to its implication in cell migration and adhesion (*Liu et al., 2010*). Mechanistically, Tgif1 exerts control over *Pak3* expression through transcriptional repression. Thus, elevated PAK3 levels contribute to the observed defects in osteoblast spreading of Tgif1-deficient cells. By using translational approaches, we demonstrate that in the absence of Tgif1, the activation of bone surfaces by osteoblasts in response to bone repair and PTH treatment is diminished. Furthermore, we uncovered that PTH facilitates osteoblast spreading via Tgif1-PAK3 signaling.

Collectively, these findings increase the knowledge on processes governing osteoblast morphology, adherence, and migration. These novel insights are important to better understand mechanisms underlying bone remodeling and repair as well as the pharmacological effects of PTH treatment.

## Results
### Tgif1-deficiency alters osteoblast morphology, adherence and migration

Recently, we reported that in Tgif1-deficient mice the number and activity of osteoblasts are reduced, which contributes to a low turnover bone remodeling (*Saito et al., 2019*). Further histological examination revealed that bone surfaces were scarcely occupied by small and flat osteoblasts in mice bearing a germline deletion of *Tgif1* (*Figure 1A*, *Figure 1—source data 1*) or a deletion of *Tgif1* targeted to osteoblasts (*Figure 1B*) compared to control littermates. This finding suggested that Tgif1-deficient osteoblasts might, in addition to an impaired bone matrix-producing capacity (*Saito et al., 2019*) be compromised in acquiring a physiological osteoblast morphology. This deficit could contribute to the low turnover bone phenotype of Tgif1-deficient mice.

Inactive and early-stage osteoblasts are rather small and flat in their morphological appearance but enlarge and adopt a more cuboidal shape upon activation. Active osteoblasts produce an extracellular matrix and have the capacity to migrate to sites of bone remodeling and repair (*Dirckx et al., 2013*). These features require the ability to adhere to surfaces and to change cell morphology to facilitate cell migration. In support of our hypothesis that Tgif1-deficiency compromises morphological plasticity of osteoblasts, calvarial- and long bone-derived osteoblasts obtained from Tgif1-deficient mice were impaired in their ability to attach to collagen type I-coated surfaces throughout a 4 hr time-course (*Figure 1C–E*). To determine the ability of Tgif1-deficient osteoblasts to migrate, we performed live cell video microscopy of osteoblasts seeded on collagen type I-coated surfaces (*Dang and Gautreau, 2018*). This analysis revealed that osteoblasts lacking Tgif1 were impaired in their migration capacity reflected by a shorter track length, a reduced velocity, and a less straight and, therefore, more meandering track path (*Figure 1F–H*, *Figure 1—figure supplement 1*). Thus, these in vivo and in vitro data indicate that Tgif1 is important for osteoblasts to adhere and migrate on collagen type I-rich surfaces. To confirm these findings in a three-dimensional functional context, we obtained long bone osteoblasts from Tgif1-defcient mice and control littermates and seeded them on a collagen type I matrix placed over a porous membrane (*Figure 1I*). In support of the previous observations, osteoblasts

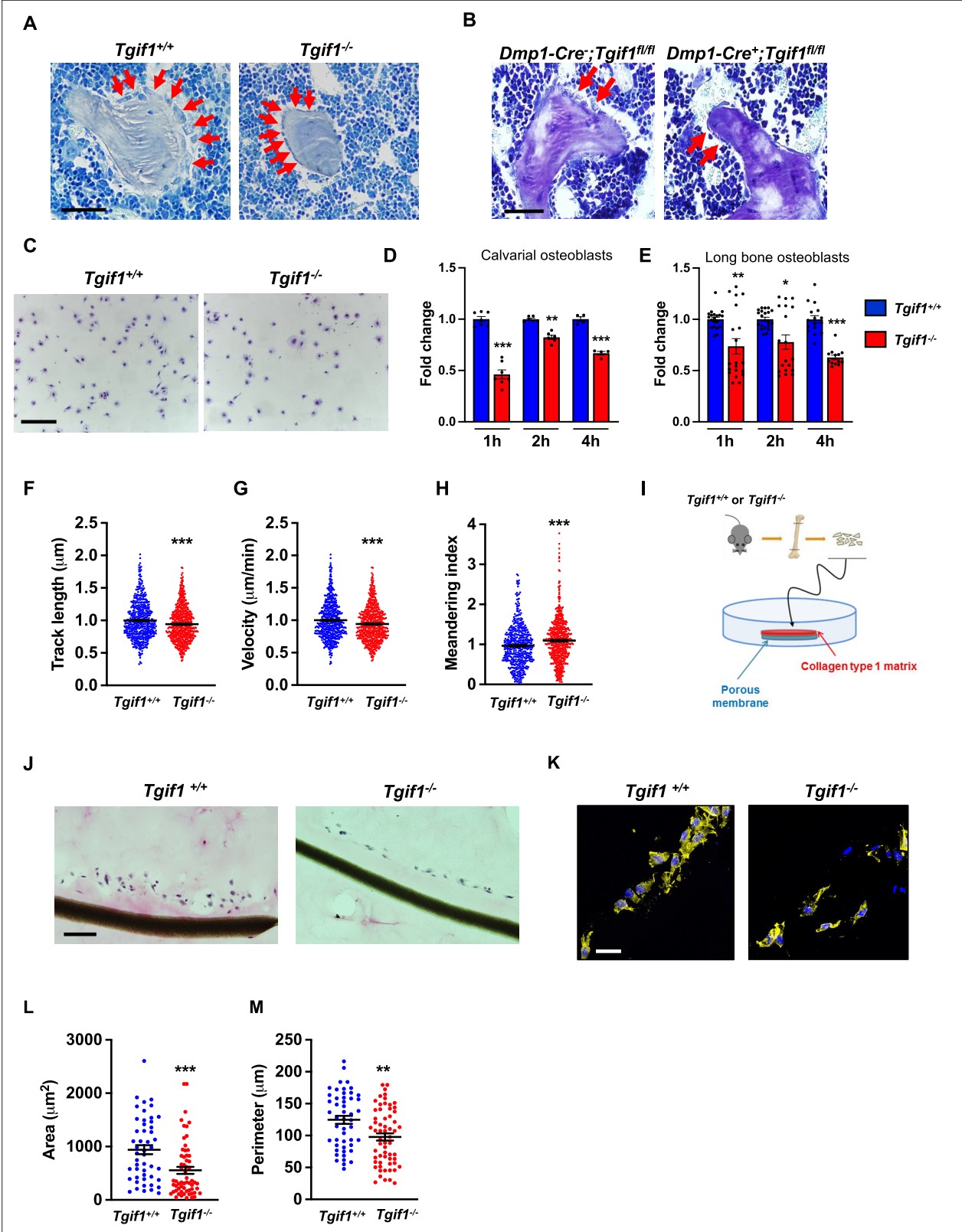

**Figure 1.** Loss of TG-interacting factor 1 (Tgif1) reduces osteoblast size in vivo and impairs osteoblast adhesion and migration in vitro. (**A**) Representative images of the femora from 12-week-old *Tgif1*⁺/⁺ (n=4) and *Tgif1*⁻/⁻ (n=4) mice stained with Toluidine blue. (**B**) Femora of 12-week-old *Dmp1-Cre⁻;Tgif1*^fl/fl^ (n=4) and *Dmp1-Cre⁺;Tgif1*^fl/fl^ (n=4) mice stained with toluidine blue. (**A, B**) Arrows indicate osteoblasts. Scale bars indicate 100 µM. (**C, D**) Osteoblasts isolated from calvariae of neonatal *Tgif1*⁺/⁺ and *Tgif1*⁻/⁻ mice upon adherence on Col-I coated surfaces for 1, 2, and 4 hr, fixation, and

*Figure 1 continued*

staining with toluidine blue. (**C**) Representative images of Toluide blue-stained cells after 2 hr of adhesion. (**D**) Quantification of adherent *Tgif1⁺/⁺* and *Tgif1⁻/⁻* calvarial osteoblasts after 1, 2, 4 hr of adhesion on Col-I coated surfaces. (**E**) Osteoblasts isolated from long bones of 8-week-old *Tgif1⁺/⁺* and *Tgif1⁻/⁻* mice upon adherence on Col-I coated surfaces for 1, 2, and 4 hr, fixation, and staining with toluidine blue. Quantification of adherent cells at indicated time points. (**F–H**) Migration of calvarial osteoblasts obtained from neonatal *Tgif1⁺/⁺* and *Tgif1⁻/⁻* mice was analyzed using live cell imaging. Quantification of (**F**) track length, (**G**) migration velocity, and (**H**) meandering index of *Tgif1⁺/⁺* and *Tgif1⁻/⁻* calvarial osteoblasts. (**I**) Long bones were harvested from *Tgif1⁺/⁺* and *Tgif1⁻/⁻* mice. Osteoblasts were isolated from bone chips and spread on Col-I matrices placed on porous membranes for 48 hr. Membranes were frozen, cut, and stained with (**J**) H&E or (**K**) phalloidin (yellow) and DAPI (blue). Quantification of cell area (**L**) and cell perimeter (**M**) of *Tgif1⁺/⁺* and *Tgif1⁻/⁻* long bone osteoblasts on Col-I matrices. Scale bars indicate 50 μm. n=minimum of three independent experiments with technical duplicates. Data are presented as mean ± SEM. Two-tailed Student's *t* test was used to compare two groups, *p<0.05, **p<0.01, ***p<0.001vs. *Tgif1⁺/⁺*.

The online version of this article includes the following source data and figure supplement(s) for figure 1:

**Source data 1.** Numerical data related to *Figure 1D, E, F, G and H*.

**Figure supplement 1.** Migration of calvarial osteoblasts obtained from neonatal *Tgif1⁺/⁺* (n=20) and *Tgif1⁻/⁻* (n=20) mice was analyzed using live cell imaging.

lacking Tgif1 adhered less to collagen type I matrix and were impaired in their capacities to migrate into the matrix compared to control cells (*Figure 1J*). The experiment also confirmed that Tgif1-deficient osteoblasts were rather flat in morphology and smaller in size as determined by reduced cell area and perimeter (*Figure 1K–M*). Collectively, these observations demonstrate that Tgif1 is indispensable for osteoblasts to adapt a regular cell morphology and to adhere and migrate on collagen type I-rich matrices in vitro and on bone surfaces in vivo.

## Tgif1-deficient osteoblasts are impaired in spreading and in forming focal adhesions

Adherence on osseous matrices and migration alongside bone surfaces towards sites of remodeling or repair are crucial functional features of osteoblasts (*Dirckx et al., 2013*; *Thiel et al., 2018*). Upon adherence and migration, osteoblasts undergo morphological changes and spread on bone surfaces, thereby decreasing their sphericity, a measure of how closely an object resembles the round shape of a sphere (*Cruz-Matías et al., 2019*). To determine the ability of Tgif1-deficient osteoblasts to spread, calvarial osteoblasts were isolated from *Tgif1⁻/⁻* mice and control littermates and subject to spreading on collagen type I-coated slides for 60 min in the presence of Calcein-AM, a dye that is fluorescent in living cells (*Dejaeger et al., 2017*). Compared to control cells, *Tgif1⁻/⁻* osteoblasts were significantly impaired in their ability to spread during the experiment (*Figure 2A and B*, *Figure 2—source data 1*).

Cell spreading is associated with the assembly and disassembly of actin filaments to form cell protrusions. Cell protrusions facilitate anchoring to the extracellular matrix via focal adhesion complexes (*Cronin and DeMali, 2021*). To determine whether Tgif1 is implicated in actin filament- and focal adhesion assembly, *Tgif1⁻/⁻* and control osteoblasts were stained for phalloidin to visualize actin filaments and Paxillin, a major component of focal adhesions. Consistently, after 60 min of culture on collagen type I-coated surfaces, Tgif1-deficient osteoblasts were less spread and, therefore, had a greater sphericity (*Figure 2C and D*). Furthermore, Tgif1-deficient osteoblasts that underwent spreading formed less cellular processes and focal adhesions compared to control cells (*Figure 2C and E*).

To confirm the findings obtained from osteoblasts bearing a germline deletion of *Tgif1*, we transiently silenced *Tgif1* in cells of the OCY454 cell line, a cell model system resembling motile osteocytes and mature osteoblasts (*Spatz et al., 2015*), using siRNA (*Figure 2—figure supplement 1*, *Figure 2—figure supplement 1—source data 1 and 2*). Consistent with the findings made by genetic deletion of *Tgif1*, siRNA-mediated silencing of *Tgif1* in OCY454 cells resulted in an impaired cell spreading with a consecutive higher cell sphericity and a decrease in focal adhesion formation (*Figure 2F–H*).

## PAK3 expression is increased in the absence of Tgif1 and impairs focal adhesion formation and osteoblast spreading

To investigate the molecular mechanisms underlying the impaired ability of *Tgif1⁻/⁻* osteoblasts to undergo morphological changes, expression, and activation of the major focal adhesion components including FAK, Integrin β1, Src, talin, paxillin, p38, and LRG5 was quantified during cell spreading. However, no major changes were identified in the expression or activation of these molecules in the

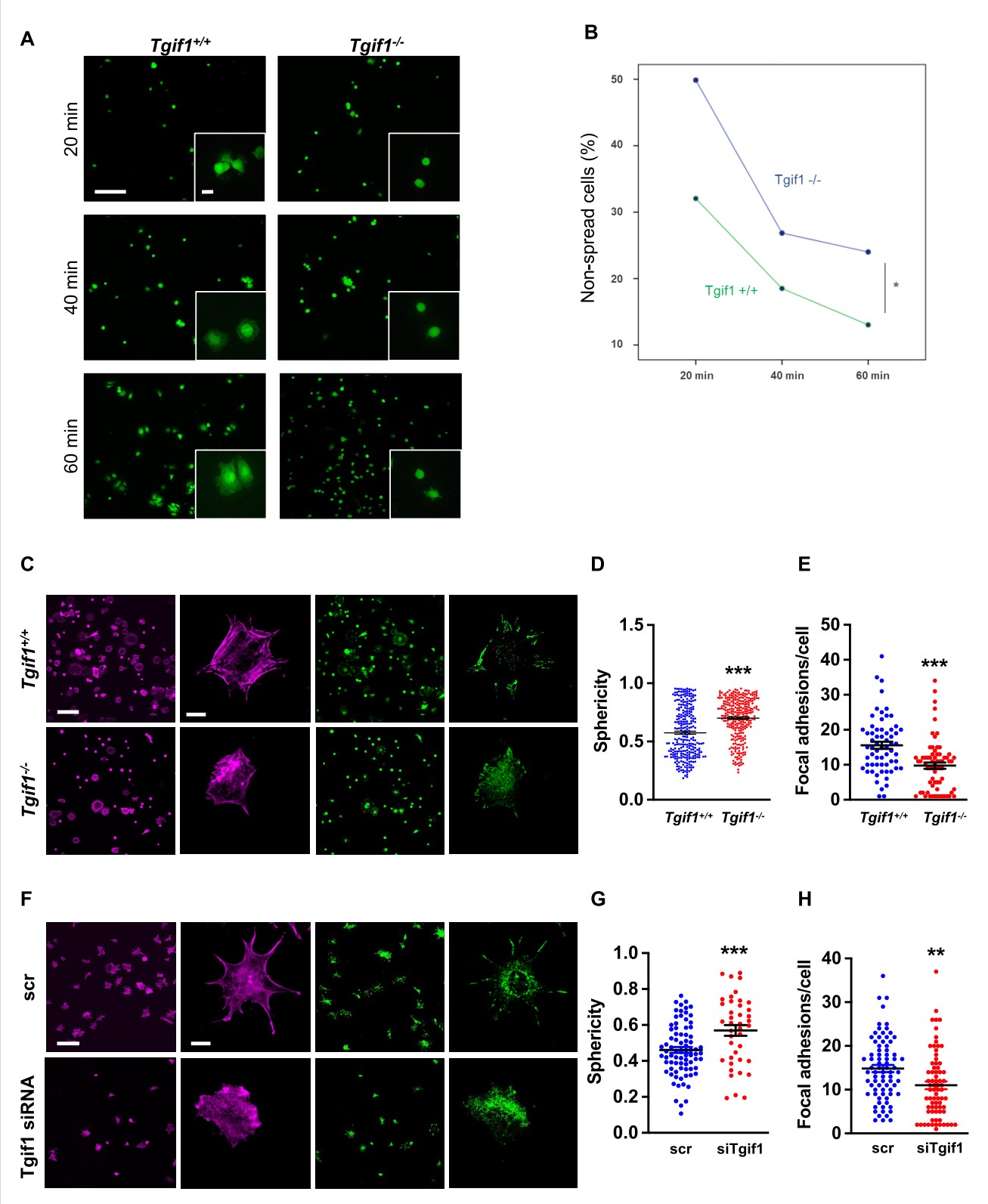

**Figure 2.** TG-interacting factor 1 (Tgif1)-deficient osteoblasts are impaired to spread and form focal adhesions. (**A, B**) Calvarial osteoblast obtained from *Tgif1*+/+ and *Tgif1*-/- mice adhered on Col-I coated slides for 20, 40, and 60 min. (**A**) Cell spreading was visualized by Calcein-AM. (**B**) The number of round (non-spread) cells was counted. Scale bar indicates 100 μm in low magnification images and 20 μm in the insets. n=6 independent experiments; Repeated Measures ANOVA, Estimated margins means test, *p<0.05 between genotypes. (**C**) *Tgif1*+/+ and *Tgif1*-/- calvarial osteoblasts were allowed

*Figure 2 continued on next page*

*Figure 2 continued*

to adhere on Col-I coated slides for 60 min. Focal adhesion formation was visualized by paxillin staining (green) and actin cytoskeleton by phalloidin (magenta). Scale bars indicate 100µm (lower magnification) or 10µm (single cells). (**D**) Analysis of single-cell sphericity after 3D reconstruction using IMARIS and (**E**) quantification of focal adhesions using Image J. (**F–H**) *Tgif1* was silenced in OCY454 cells using siRNA and cells were allowed to adhere on Col-I coated slides for 60 min. (**F**) Focal adhesion formation was visualized by Paxillin staining (green) and cell protrusions by phalloidin (magenta). Scale bars indicate 100µm (lower magnification) or 10µm (single cells). (**G**) Analysis of cell sphericity after 3D reconstruction and (**H**) quantification of focal adhesions. n=6 independent experiments; Data are presented as mean±SEM. Unpaired t-test, \*\*\*p<0.001, \*\*p<0.01 vs. *Tgif1⁺/⁺* (**D, E**), and scr (**G, H**).

The online version of this article includes the following source data and figure supplement(s) for figure 2:

**Source data 1.** Numerical data related to *Figure 2D, E, G and H*.

**Figure supplement 1.** Representative image of an immunoblot demonstrating the expression of TG-interacting factor 1 (Tgif1) in OCY454 cells transfected with scrambled control siRNA (Scr) or siRNA against *Tgif1* (siTgif1).

**Figure supplement 1—source data 1.** Original files for the western blot analysis in *Figure 2—figure supplement 1* (anti-Tgif1 and anti-actin).

**Figure supplement 1—source data 2.** PDF containing *Figure 2—figure supplement 1A* and original scans of the relevant western blot analysis (anti-Tgif1 and anti-actin) with highlighted bands and sample labels.

absence of Tgif1 (*Figure 3—figure supplement 1A*, *Figure 3—figure supplement 1—source data 1 and 2*).

Next, we examined the localization of Tgif1 to determine a possible interaction of Tgif1 with components of focal adhesions or actin filaments. Immunocytochemistry revealed that Tgif1 is predominantly localized in the nucleus and to some extend in the cytoplasm (*Figure 3—figure supplement 1B*). However, cytoplasmic Tgif1 did not interact with focal adhesions, because no co-localization of eGFP-Tgif1 neither with paxillin nor with talin (*Figure 3—figure supplement 1B*) was observed.

Given the predominantly nuclear localization of Tgif1 (*Figure 3—figure supplement 1B*) and its established function as a transcriptional repressor (*Saito et al., 2019*; *Wotton et al., 1999*), we investigated whether absence of Tgif1 might alter the expression of the genes involved in actin cytoskeletal assembly and re-arrangement as a prerequisite of cell adhesion and migration. First, we quantified the expression of members of the cell division control protein 42 homolog (Cdc42), which are small GTPases of the Rho family that participate in the control of multiple cellular functions including cell migration and cell morphology (*Hirsch et al., 2001*). However, expression of none of the Cdc42 family members *Cdc42es2*, *Cdc42ep1*, *Cdc42ep2*, or *Cdc42ep4* was changed in the absence of Tgif1 during osteoblast adhesion and spreading (*Figure 3—figure supplement 2A–D*, *Figure 3—figure supplement 2—source data 1*). These data indicate that the impaired spreading of Tgif1-deficient osteoblasts is unlikely caused by Cdc42 family members and that other factors are implicated in this process. We, therefore, quantified the expression of p21-activated-kinase (*Pak*) family members, who have been shown to play a role in cell adhesion and migration as well as in actin nucleation in neurons (*Liu et al., 2010*; *Kreis and Barnier, 2009*). While expression of *Pak1, Pak2*, and *Pak4* was unchanged in Tgif1-deficient osteoblasts during adhesion and spreading (*Figure 3—figure supplement 2E–G*), expression of *Pak3* was strongly increased in Tgif1-deficient osteoblasts compared to control cells at the mRNA and protein level prior to and 60 min after adhesion (*Figure 3A and B*, *Figure 3—source data 1–3*). Confirming this observation in vivo, expression of *Pak3* mRNA showed a trend towards an increase in bones from *Tgif1⁻/⁻* mice compared to control littermates (*Figure 3C*). These findings suggest that the impaired adhesion and spreading of Tgif1-deficient osteoblasts might be related to a deregulated abundance of *Pak3*.

To further investigate the regulation of *Pak3* expression by Tgif1, we performed an in silico analysis of the *Pak3* promoter and identified 3 putative Tgif1 binding sites (*Figure 3—figure supplement 3A*). Since only one site was species-conserved between rat and mouse, we verified the binding of Tgif1 to this site using chromatin immunoprecipitation. Indeed, Tgif1 bound to the predicted promoter binding site and to a site of the *Rar* promoter as positive control (*Zhang et al., 2009*), but not to a DNA sequence lacking predicted Tgif1 binding site as negative control (*Figure 3D*), suggesting a regulation of the *Pak3* promoter activity by Tgif1. To address this question, *Tgif1* expression was silenced in OCY454 osteoblast-like cells using siRNA followed by transfection of the cells with a 2.3 kb fragment of the rat *Pak3* promoter upstream of the luciferase gene. Silencing of *Tgif1* lead to a significant increase in luciferase activity compared to control (*Figure 3E*) demonstrating that Tgif1 is a transcriptional repressor of the *Pak3* promoter.

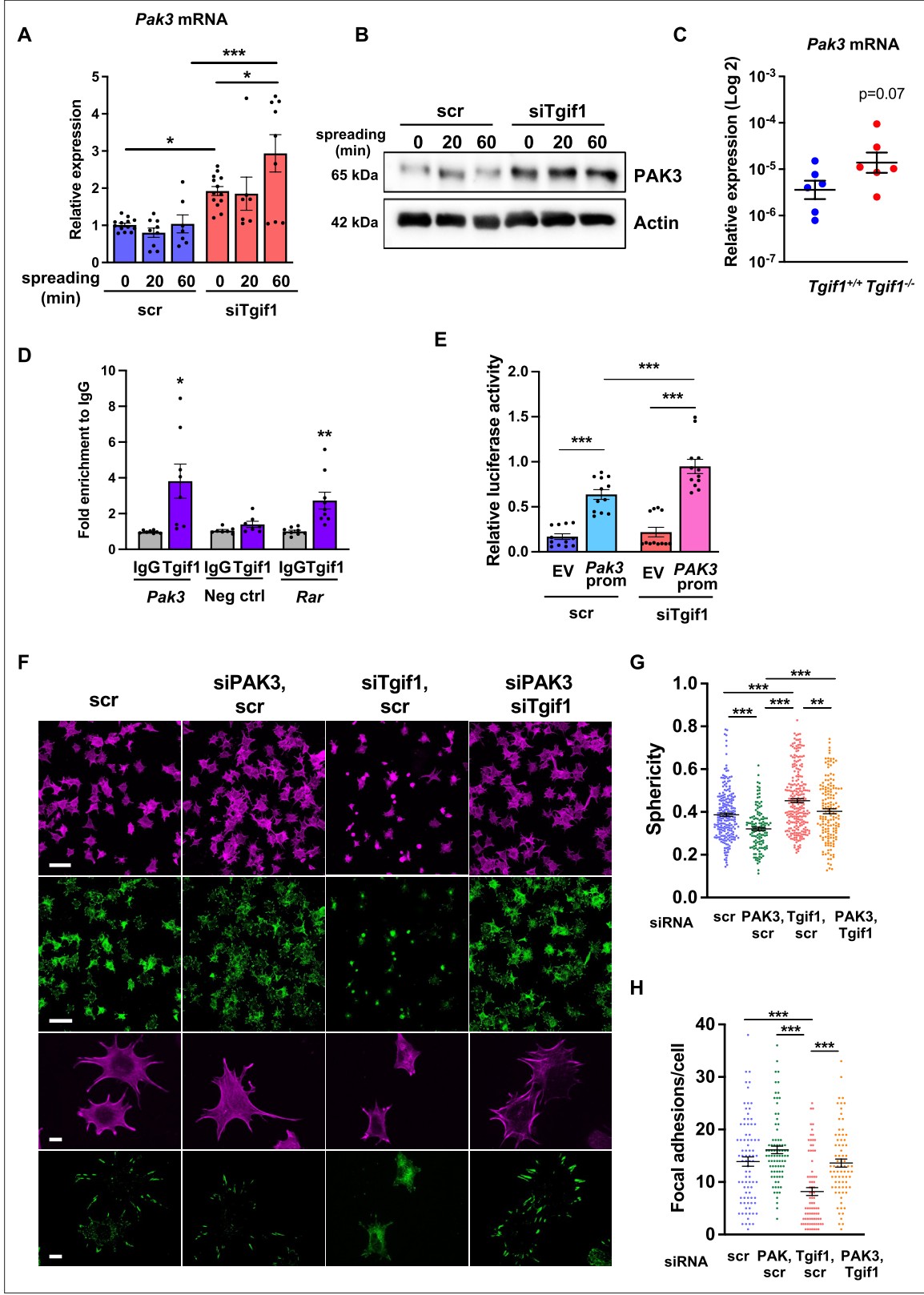

**Figure 3.** Silencing of p21-activated kinase 3 (PAK3) restores cell spreading and focal adhesion formation in TG-interacting factor 1 (Tgif1)-deficient cells. (**A**) *Pak3* mRNA expression in OCY454 cells transfected with siRNA targeting *Tgif1* (siTgif1) or scrambled control (scr) siRNA before (0) and after 20 and 60 min of spreading on Col-I coated slides. n=9, 1-way ANOVA with Tukey's multiple comparison test, *p<0.05, ***p<0.001. (**B**) Representative images of immunoblots demonstrating the abundance of PAK3 expression upon silencing of Tgif1. Actin was used as a control. (**C**) *Pak3* mRNA

*Figure 3 continued on next page*

*Figure 3 continued*

expression in tibiae of 12-week-old *Tgif1*[+/+] (n=6) and *Tgif1*[-/-] (n=6) mice. (**D**) Tgif1 binding to the predicted site of the *Pak3* promoter in OCY454 cells analyzed by ChIP and quantified as fold enrichment to the relative IgG control. Negative and positive (*Rarα*) controls were used as indicated. n=8, unpaired t-test with Welch's correction, *p<0.05, **p<0.01 vs. respective IgG control. (**E**) Tgif1-deficient (siTgif1) or control (scr) OCY454 cells were transfected with renilla plasmid and a pGL3 plasmid (EV) or a pGL3 plasmid containing a 2.3 kb fragment of the rat *Pak3* promoter upstream of the luciferase gene. The promoter activity was quantified using luciferase assays and presented as relative luciferase activity (luciferase/renilla). One-way ANOVA, Tukey's multiple comparison test, ***p<0.001. (**F–H**) OCY454 cells were transfected alone or in combinations with siTgif1 for 48 hr, siPak3 for 24 hr, and scrambled (scr) control. (**F**) Cells were allowed to adhere on Col-I coated slides for 60 min. Formation of focal adhesions was visualized by paxillin staining (green) and actin cytoskeleton by phalloidin staining (magenta). Scale bars indicate 100μm (two upper rows) or 10 μm (two lower rows). (**G**) Quantification of cell sphericity using IMARIS. (**H**) Quantification of the number of mature focal adhesions per cell using the Image J software. n=4 independent experiments in which individual cells were analyzed. One-way ANOVA, Tukey's multiple comparison test, **p<0.01 ***p<0.001. Data are presented as mean ± SEM.

The online version of this article includes the following source data and figure supplement(s) for figure 3:

**Source data 1.** Numerical data related to *Figure 3A, C, D, E, G and H*.

**Source data 2.** Original files for the western blot analysis in *Figure 3B* (anti-PAK3 and anti-actin).

**Source data 3.** PDF containing *Figure 3B* and original scans of the relevant western blot analysis (anti-PAK3 and anti-actin) with highlighted bands and sample labels.

**Figure supplement 1.** TG-interacting factor 1 (Tgif1)-deficiency does not alter the abundance or activation of the FA components and Tgif1 does not co-localize with FA complexes.

**Figure supplement 1—source data 1.** Original file for the western blot analysis in *Figure 3—figure supplement 1A* (anti-pFAK, anti-FAK, anti-pp38, anti-p38, anti-ppaxillin, anti-paxillin, anti-psrc, anti-src, anti-integrin β1, anti LRG5, anti-talin, anti-actin).

**Figure supplement 1—source data 2.** PDF containing *Figure 3—figure supplement 1A* and original scans of the relevant western blot analysis (anti-pFAK, anti-FAK, anti-pp38, anti-p38, anti-ppaxillin, anti-paxillin, anti-psrc, anti-src, anti-integrin β1, anti LRG5, anti-talin, anti-actin) with highlighted bands and sample labels.

**Figure supplement 2.** Lack of TG-interacting factor 1 (Tgif1) does not alter the expression of *Cdcs*, *Pak1*1, *Pak2*, or *Pak4*.

**Figure supplement 2—source data 1.** Numerical data related to *Figure 3—figure supplement 2A-G*.

**Figure supplement 3.** Tgif1 binding site on the PAK3 promoter and expression of PAK3 and Tgif1.

**Figure supplement 3—source data 1.** Original file for the western blot analysis in *Figure 3—figure supplement 3* (anti-PAK3, anti-Tgif1, and anti-actin).

**Figure supplement 3—source data 2.** PDF containing *Figure 3—figure supplement 3* and original scans of the relevant western blot analysis (anti-PAK3, anti-Tgif1, and anti-actin) with highlighted bands and sample labels.

To determine if an elevated abundance of Pak3 due to Tgif1-deficiency impairs osteoblast spreading, *Pak3* expression was silenced in Tgif1-deficient OCY454 cells using siRNA (*Figure 3— figure supplement 3B*, *Figure 3—figure supplement 3—source data 1 and 2*). Indeed, attenuating *Pak3* expression restored the *Tgif1* deficiency-mediated impaired cell spreading (*Figure 3F*). Since cells are less spherical upon spreading, *Pak3* silencing reduced the *Tgif1* deficiency-dependent gain in cell sphericity and restored the number of focal adhesions per cell that were reduced by *Tgif1* deficiency to the level of control (*Figure 3G and H*). Together, these findings demonstrate that in the absence of Tgif1, the *Pak3* promoter is de-repressed, leading to an increased transcriptional activity and *Pak3* expression. A higher abundance of Pak3 in turn suppresses focal adhesion formation and osteoblast spreading.

## Tgif1 expression increases during osteoblast spreading via activation of ERK1/2 and AP1 signaling pathways

To further investigate the role of Tgif1 during osteoblast spreading, we quantified *Tgif1* expression during this process. Both, *Tgif1* mRNA and protein expression were significantly increased within 20 min of spreading with a further increase after 60 min in calvarial osteoblasts (*Figure 4A and B*, *Figure 4—source data 1*) and in OCY454 cells (*Figure 4C and D*). To determine if the increase in Tgif1 abundance is transcriptionally regulated, osteoblasts were treated with the transcription inhibitor Actinomycin-D during spreading, which fully prevented the increase in *Tgif1* mRNA expression (*Figure 4E*).

Since binding of cells to collagen I-coated surfaces activates the ERK1/2 pathway (*Emerson et al., 2009*; *Fincham et al., 2000*), we investigated the potential implication of ERK1/2 signaling in the

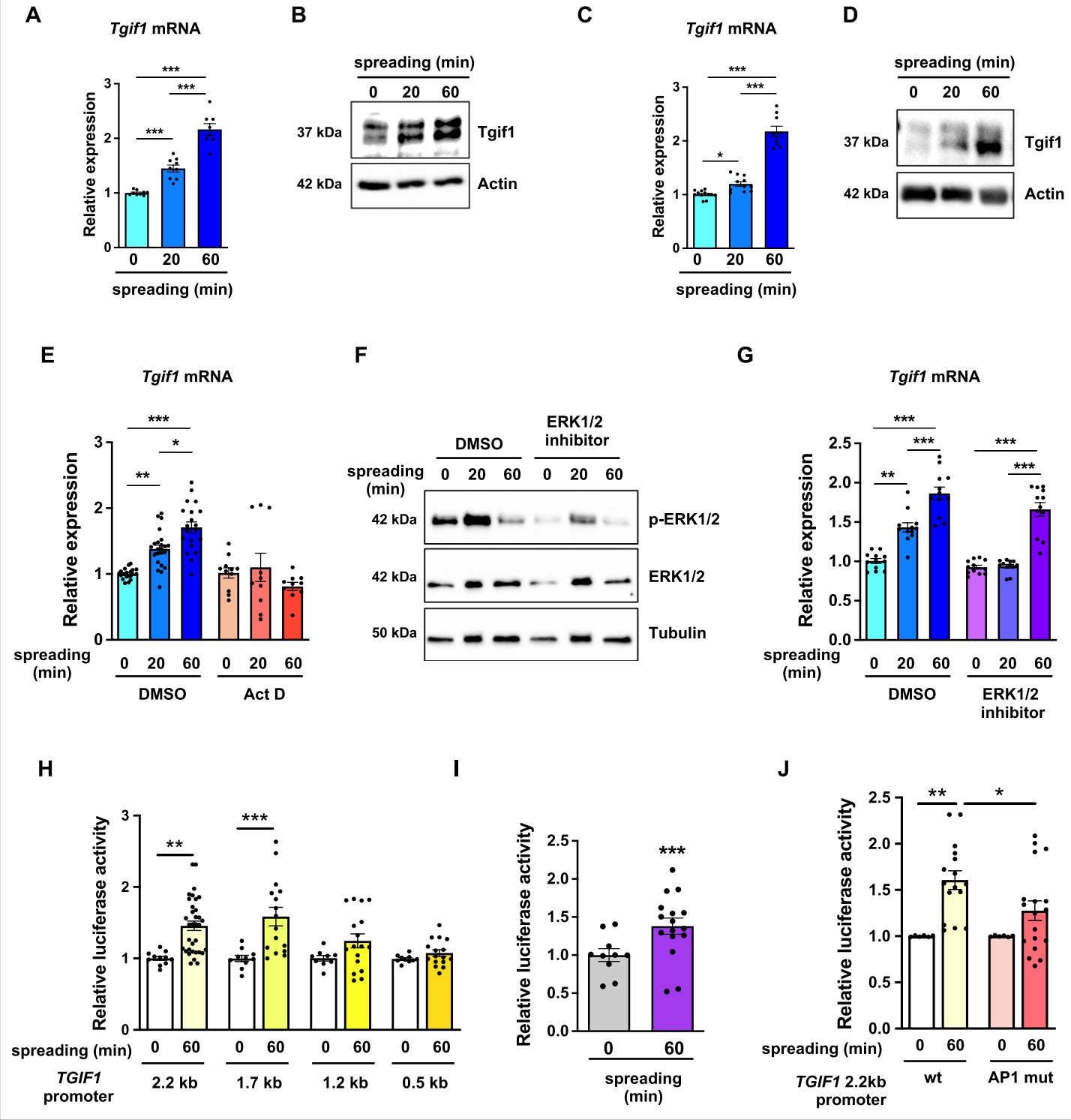

**Figure 4.** TG-interacting factor 1 (Tgif1) expression is increased during cell spreading via ERK and AP1 signaling pathways. (**A**) Quantification of *Tgif1* mRNA expression and (**B**) representative images of immunoblots of Tgif1 expression in calvarial osteoblasts before (0 min) and after 20 and 60 min of spreading. Actin was used as a control. (**C**) *Tgif1* mRNA expression and (**D**) representative images of immunoblots of Tgif1 expression in OCY454 cells before (0 min) and after 20 and 60 min spreading on Col-I coated slides. Actin was used as a control. (**E**) OCY454 cells were treated with DMSO as control or with 5 µM Actinomycin D for 15 min prior to adherence on Col-I coated slides for 20 and 60 min. *Tgif1* mRNA was quantified before (0 min) and during spreading. (**F**) Representative immunoblot images demonstrating the efficiency of ERK1/2 inhibitor SCH772984 (1 µg/ml) to prevent ERK1/2 phosphorylation (n=4). (**G**) Quantification of *Tgif1* mRNA expression during spreading after 15 min pre-treatment with DMSO (control) or with the ERK

*Figure 4 continued on next page*

*Figure 4 continued*

inhibitor SCH772984. (**H**) Quantification of the *TGIF1* promoter activity during cell spreading. OCY454 cells were transfected with a luciferase reporter plasmid encoding a 2.2 kb fragment of the *TGIF1* promoter or progressive truncations thereof (1.7, 1.2, and 0.5 kb) along with a plasmid encoding renilla firefly as control. Upon spreading for 60 min, promoter activity was quantified using a dual luciferase reporter gene assay and presented as normalized luciferase activity (luciferase/renilla). (**I**) Quantification of the AP1 transcriptional activity during cell spreading. OCY454 cells were transfected with a 6X-TRE-luciferase reporter to determine AP-1 activity along with a plasmid encoding renilla firefly as control. Upon spreading for 60 min, promoter activity was quantified using a dual luciferase reporter gene assay and presented as normalized luciferase activity (luciferase/renilla). (**J**) Quantification of the activity of the wild-type (wt) *TGIF1* 2.2 kb promoter and of the same fragment bearing a mutant AP1 binding site (AP1 mut) during cell spreading. OCY454 cells were transfected either with a luciferase reporter plasmid encoding a wild-type (wt) 2.2 kb fragment of the *TGIF1* promoter or with the same promoter in which the AP1 binding site has been mutated (mut) along with a plasmid encoding renilla firefly as control. Upon spreading for 60 min, the promoter activity was quantified using a dual luciferase reporter gene assay and presented as normalized luciferase activity (luciferase/renilla). n = minimum 4 independent experiments. Data are presented as mean ± SEM. Two-tailed Student's *t* test was used to compare two groups (I) One-way ANOVA, Tukey's multiple comparisons test was used to compare multiple groups, *$p<0.05$, **$p<0.01$, ***$p<0.001$ vs. respective control.

The online version of this article includes the following source data for figure 4:

**Source data 1.** Numerical data related to *Figure 4A, C, E, G, H, I and J*.

**Source data 2.** Original files for the western blot analysis in *Figure 4B* (anti-Tgif1 and anti-actin).

**Source data 3.** PDF containing *Figure 4B* and original scans of the relevant western blot analysis (anti-Tgif1 and anti-actin) with highlighted bands and sample labels.

**Source data 4.** Original files for the western blot analysis in *Figure 4D* (anti-Tgif1 and anti-actin), anti-pERK1/2, anti-ERK1/2, and anti-tubulin.

**Source data 5.** PDF containing *Figure 4D and F* and original scans of the relevant western blot analysis (anti-Tgif1 and anti-actin) with highlighted bands and sample labels.

**Source data 6.** Original files for the western blot analysis in *Figure 4F* (anti-pERK1/2, anti-ERK1/2, and anti-tubulin).

**Source data 7.** PDF containing *Figure 4F* and original scans of the relevant western blot analysis (anti-pERK1/2, anti-ERK1/2, and anti-tubulin) with highlighted bands and sample labels.

increase of *Tgif1* gene expression during osteoblast spreading. Experimentally, osteoblasts were treated with an ERK1/2 inhibitor, which greatly attenuated ERK1/2 phosphorylation after 20 min and 60 min of osteoblast spreading (*Figure 4F*). Although ERK1/2 inhibition prevented the increase of *Tgif1* mRNA expression during the first 20 min of osteoblast spreading, it failed to suppress *Tgif1* mRNA expression 60 min after initiation of osteoblast spreading (*Figure 4G*). This finding suggests that the increase in *Tgif1* mRNA expression at early stages of osteoblast spreading is mediated by ERK1/2 signaling while it is independent of ERK1/2 signaling at later stages of osteoblast spreading.

To elucidate the molecular mechanisms underlying the activation of *Tgif1* transcription at later stages of osteoblast spreading, we performed progressive truncations of the 2.2 kb fragment of the human *TGIF1* promoter (*Saito et al., 2019*). While the 2.2 kb and 1.7 kb fragments of the human *TGIF1* promoter were fully activated after 60 min of osteoblast spreading, no activation of the 1.2 kb and 0.5 kb fragments was observed (*Figure 4H*), suggesting that the regulatory promoter region that is activated during osteoblast spreading must be located within the 500 bp between the 1.7 kb and the 1.2 kb promoter fragments.

Recently we reported the presence of an AP1 binding site in this region of the *TGIF1* promoter (*Saito et al., 2019*), suggesting that AP1 signaling might be implicated in this process. Since the role of AP1 signaling in the context of cell spreading has not been fully elucidated, we first determined the activation of an AP1-responsive promoter element during osteoblast spreading. The findings demonstrate that AP1-signaling is activated 60 min after initiation of osteoblast spreading (*Figure 4I*), indicating a potential implication of AP1 signaling during advanced stages of osteoblast spreading. To further test this hypothesis, osteoblasts were transfected either with a plasmid encoding the wild-type *TGIF1* promoter bearing an AP1 binding site or with a plasmid in which the AP1 binding site was disabled by site-specific mutation (*Saito et al., 2019*). Although the wild-type promoter was fully activated after 60 min of osteoblast spreading, the *TGIF1* promoter bearing the disabled AP1 binding site was not activated (*Figure 4J*). These findings suggest that AP1 signaling is involved in the activation of *Tgif1* gene expression during later stages of osteoblast spreading.

## Tgif1 deficiency leads to fewer and less active osteoblasts during bone regeneration

Bone has a regenerative capacity to heal upon fracture. In fulfilling this function, osteoblast adherence and spreading are crucial features since osteoblasts migrate to sites of bone repair for subsequent matrix production and regeneration (*Thiel et al., 2018*; *Dirckx et al., 2013*). Fracture healing involves various cell types and cellular processes (*Einhorn and Gerstenfeld, 2015*). One important component is the activation of periosteal cells to migrate alongside vessels that sprout into the fracture zone (*Maes et al., 2010*). Periosteal cells become active osteoblasts that produce a cartilage template, which then ossifies and forms new trabeculae to consolidate the fracture gap (*Duchamp de Lageneste et al., 2018*). Since this process involves adherence and spreading of osteoblasts, we hypothesized that Tgif1 might be important in the context of bone repair. To test this hypothesis, we performed open mid-shaft tibia fractures in 10-weeks-old adult male mice bearing a germline deletion of *Tgif1* and control littermates, followed by fracture stabilization using an intramedullary pin. Three weeks after fracture, newly formed trabeculae were densely occupied by cuboidal matrix-producing osteoblasts in control animals (*Figure 5A–E*, *Figure 5—source data 1*). In contrast, in mice bearing a germline deletion of *Tgif1*, the callus was less ossified and the trabeculae in the repair zone were sparsely occupied by fewer, thinner, and less-active osteoblasts (*Figure 5A–F*). These findings demonstrate that Tgif1 is important for bone regeneration.

## Activation of bone surfaces by PTH is attenuated in the absence of Tgif1 in osteoblasts

Intermittent administration of PTH augments bone formation, leading to a remodeling-based increase in bone mass, bone mineral density and a consecutive decrease in fracture rate in humans (*Neer et al., 2001*; *Taipaleenmäki et al., 2022*; *Baron and Hesse, 2012*). This anabolic function requires multiple alterations at the cellular level. For instance, PTH induces the differentiation of mesenchymal precursor cells and reverts quiescent bone lining cells into active osteoblasts (*Nishida et al., 1994*; *Dobnig and Turner, 1995*; *Kim et al., 2012*). Furthermore, PTH increases the number and bone-forming capacity of mature osteoblasts (*Mizoguchi and Ono, 2021*; *Jilka et al., 2009*).

Recently, we identified *Tgif1* as a PTH target gene in osteoblasts that is necessary for the full PTH-mediated bone mass accrual (*Saito et al., 2019*). In addition, the site of the *TGIF1* promoter known to be activated by PTH signaling (*Saito et al., 2019*), is identical with the site identified here that is activated by AP1 signaling during osteoblast adhesion (*Figure 4J*). In this study, we also uncovered that Tgif1 is a physiological regulator of osteoblast adherence, spreading, and migration. Collectively, this evidence let us to propose that Tgif1 might also be implicated in the PTH-mediated bone surface activation.

To determine whether Tgif1 is implicated in the PTH-mediated activation of bone surfaces by osteoblasts, we performed histomorphometric analysis of bone sections from mice bearing a germline deletion of *Tgif1* and control littermates that were injected with PTH or vehicle control. Consistent with the findings reported by others and in support of our hypothesis, PTH treatment greatly increased the number of active osteoblasts and induces the acquisition of a cuboidal shape and consequently the percentage of active bone surfaces (*Kousteni and Bilezikian, 2008*; *Tam et al., 1982*), which all occurred at a much lesser extent in mice lacking Tgif1 (*Figure 6A and B*, *Figure 6—source data 1*). To determine whether this effect is osteoblast-autonomous, we performed histomorphometric analysis of bones obtained from mice in which *Tgif1* deletion was targeted to mature osteoblasts using the Dmp1-promoter (*Bivi et al., 2012*). Indeed, while PTH treatment induced osteoblast recruitment, adaptation of a cuboidal morphology of matrix-producing osteoblasts, and a profound activation of bone surfaces by osteoblasts in control animals, these effects were much less pronounced in mice lacking Tgif1 in osteoblasts (*Figure 6C and D*). These findings, therefore, demonstrate that PTH-dependent bone surface activation is attenuated in the absence of Tgif1 in osteoblasts.

## PTH facilitates spreading of osteoblasts via Tgif1-PAK3 signaling

The observation that Tgif1 is crucial for osteoblast adherence, spreading, and migration in vitro as well as for the PTH-mediated activation of bone surfaces suggests that PTH may facilitate the spreading of osteoblasts and, therefore, the decrease in cell sphericity via Tgif1. To test this hypothesis, OCY454 cells were treated with PTH or vehicle control during spreading. Immunofluorescence staining and

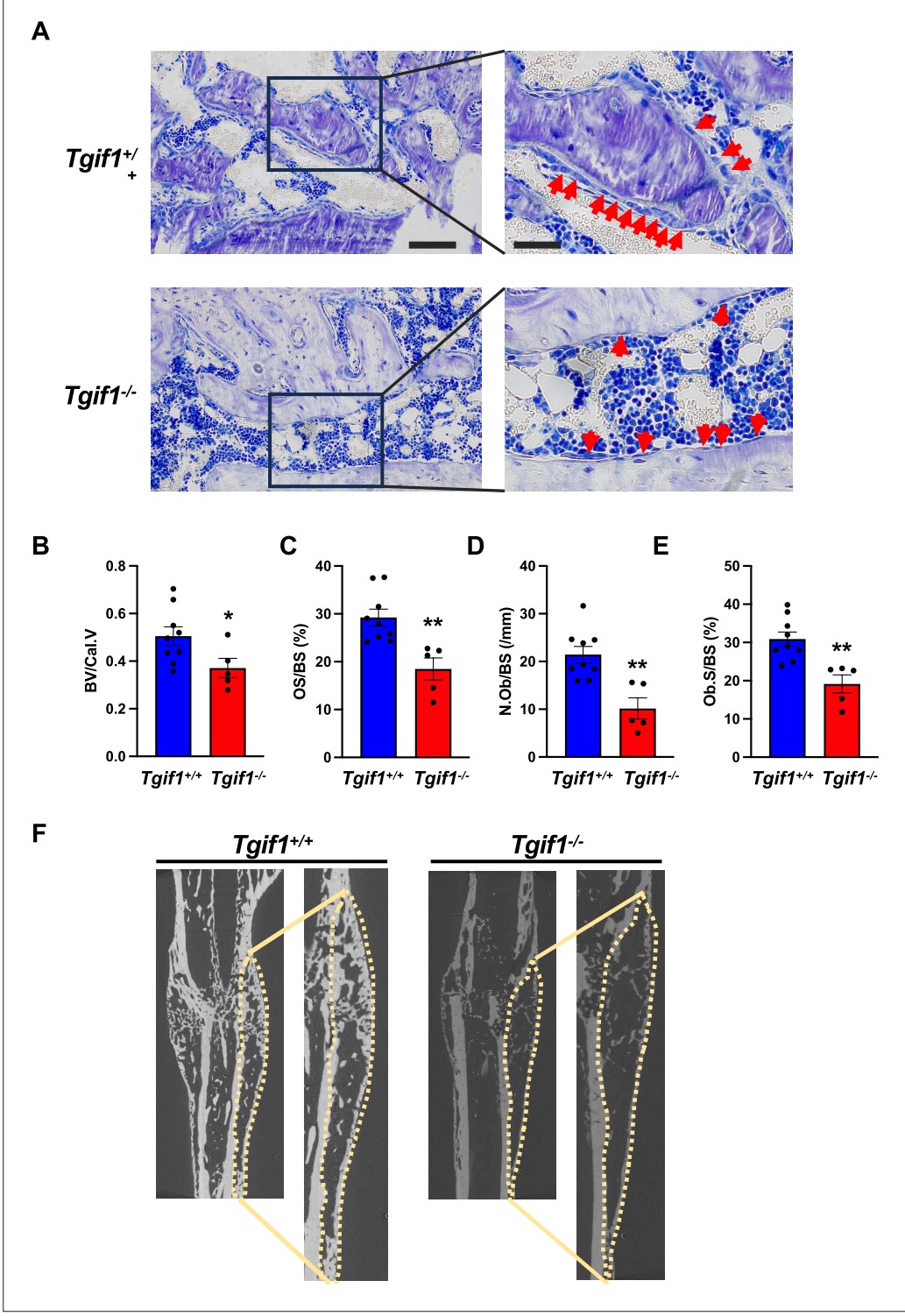

**Figure 5.** TG-interacting factor 1 (Tgif1) deficiency causes a reduced number and activity of osteoblasts during bone repair. (**A**) Representative images of Toluidine blue-stained sections of bones from male *Tgif1*+/+ (n=9) and *Tgif1*-/- mice (n=5) that received an open fracture of the midshaft tibia at 10 weeks of age followed by bone healing for 3 weeks upon intramedullary stabilization. Arrows indicate osteoblasts. Scale bars indicate 100 μm (left panels) and 50 μm (right panels). (**B - E**) Histological sections were used to quantify the histomorphometric parameters (**B**) bone volume per callus volume (BV/Cal.V), (**C**) osteoid surface per bone surface (OS/BS), (**D**) number of osteoblasts per bone surface (N.Ob/BS) and (**E**) osteoblast surface per bone surface (Ob.S/BS). (**F**) Representative micro-

*Figure 5 continued on next page*

*Figure 5 continued*

computed tomography (µCT) images of the tibiae of *Tgif1⁺/⁺* and *Tgif1⁻/⁻* mice 21 days after fracture. Data are presented as mean ± SEM. Unpaired t-test, *p<0.05, **p<0.01 vs. *Tgif1⁺/⁺*.

The online version of this article includes the following source data for figure 5:

**Source data 1.** Numerical data related to *Figure 5B, C, D and E*.

quantification of cell sphericity revealed that PTH supports osteoblast spreading (*Figure 7A and B*, *Figure 7—source data 1*). Furthermore, PTH treatment of cells in which *Tgif1* expression was silenced failed to induce the spreading of osteoblasts (*Figure 7A and B*), demonstrating that the PTH-mediated osteoblast spreading indeed depends on Tgif1.

Since *Pak3* expression is increased in the absence of Tgif1 and impairs focal adhesion formation and osteoblast spreading, we investigated the possibility that PTH decreases *Pak3* gene expression via Tgif1. Indeed, *Pak3* mRNA expression was reduced upon PTH treatment (*Figure 7C*). In addition, PTH treatment increased Tgif1 expression and reduced PAK3 protein abundance in control cells but not in osteoblasts in which *Tgif1* expression was silenced by siRNA (*Figure 7D*, *Figure 7—source data 2 and 3*). Next, we examined osteoblast spreading upon PTH treatment in the absence of Tgif1 alone or in combination with loss of PAK3. Immunofluorescence staining and quantification of cell sphericity revealed that silencing Tgif1 expression impaired osteoblast spreading induced by PTH treatment, which was restored by the concomitant silencing of *Pak3* expression (*Figure 7E and F*). Furthermore, lack of both, Tgif1 and PAK3 prevented PTH-induced decrease in cell sphericity. These findings indicate that an increased abundance of PAK3 in response to Tgif1-deficiency impairs PTH-induced spreading of osteoblasts and that PTH facilitates osteoblast spreading via Tgif1-PAK3 signaling.

In summary, our results reveal that Tgif1 suppresses *Pak3* expression in osteoblasts via the ERK1/2 and AP1 signaling pathways, which is important for osteoblasts to form focal adhesions, to spread on bone surfaces, and to migrate. This mechanism is implicated in bone regeneration and in the pharmacological effects of PTH treatment and is, therefore, of translational relevance.

## Discussion

This study revealed that Tgif1-deficient osteoblasts exhibit an altered morphology, reduced adherence to collagen type I-coated surfaces, impaired migration capacity, and decreased spreading compared to control cells. These defects in Tgif1-deficient osteoblasts are associated with compromised focal adhesion formation and increased expression of PAK3. The findings further demonstrate that Tgif1 regulates *Pak3* expression through transcriptional repression and that elevated PAK3 abundance contributes to the impaired osteoblast spreading observed in Tgif1-deficient cells. Additionally, Tgif1 is shown to participate in the activation of bone surfaces in the context of bone regeneration and PTH treatment since in the absence of Tgif1, bone regeneration- and PTH-induced activation of bone surfaces is attenuated. Mechanistically, PTH promotes osteoblast spreading via Tgif1-PAK3 signaling, with *Pak3* expression being reduced by PTH treatment in a Tgif1-dependent manner. Overall, this study emphasizes the importance of Tgif1 in regulating osteoblast morphology, adherence, and migration through the modulation of PAK3 expression, providing novel, and translationally relevant mechanistic insights into the regulation of the cytoskeletal architecture of osteoblasts.

Both, bone repair and PTH treatment are frequent and clinically relevant events. Fractures often occur in the context of high-energy accidents, however, bone fragility syndromes such as osteoporosis also cause fractures upon minimal or no trauma (*Malluche et al., 2013*; *Löfman et al., 2007*). Bone regeneration, a complex process involving osteoblast recruitment, matrix production, and tissue regeneration, is compromised in Tgif1-deficient mice. Activation of bone surfaces by osteoblasts, a critically important aspect of bone repair, is significantly reduced in the absence of Tgif1, leading to a reduced callus ossification. In addition, Tgif1-deficient mice also have a reduced number and activity of osteoclasts (*Saito et al., 2019*) and, therefore, an attenuated bone remodeling which might also contribute to an impaired bone healing. Nevertheless, the results presented here indicate that Tgif1 is important for osteoblast recruitment to the site of repair and subsequent bone surface activation, which extends the established function of Tgif1 beyond the regulation of osteoblast differentiation and activity.

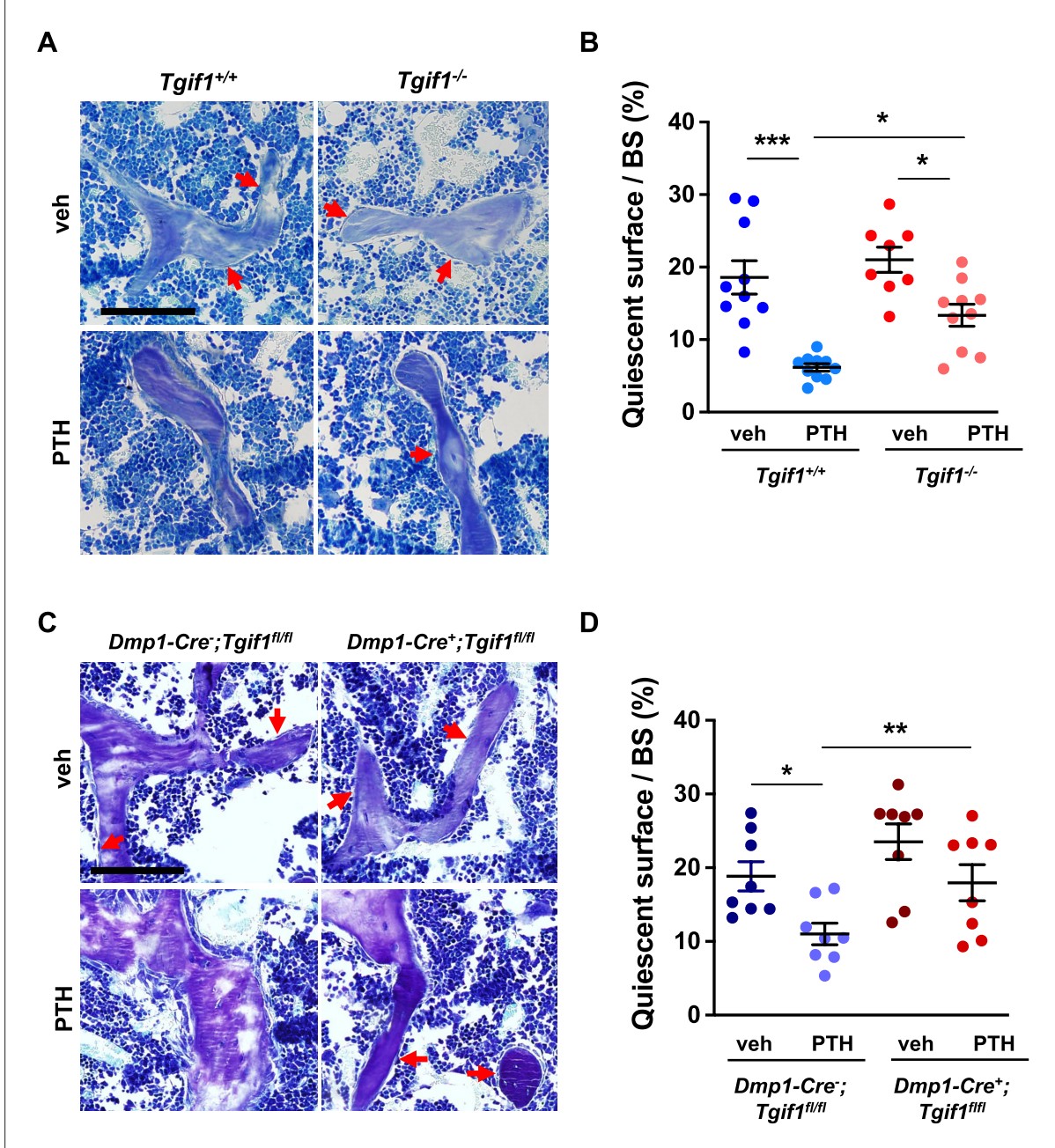

**Figure 6.** PTH is impaired in activating quiescent bone surfaces in TG-interacting factor 1 (*Tgif1*)-deficient mice. (**A**) Toluidine blue staining of tibiae from 12-week-old *Tgif1*$^{+/+}$ and *Tgif1*$^{-/-}$ mice treated with vehicle (veh) or PTH. Quiescent surfaces are indicated by red arrows. Representative images are shown. Scale bar indicates 100 µm. (**B**) Quantification of the percentage of quiescent surfaces per bone surface (BS). *Tgif1*$^{+/+}$, veh n=10; *Tgif1*$^{+/+}$, PTH n=10, *Tgif1*$^{-/-}$, veh n=8; *Tgif1*$^{-/-}$, PTH n=10. (**C**) Toluidine blue staining of tibiae from 12-week-old *Dmp1-Cre*$^{-}$;*Tgif1*$^{fl/fl}$ and *Dmp1-Cre*$^{+}$;*Tgif1*$^{fl/fl}$ mice treated with vehicle (veh) or PTH. Quiescent surfaces are indicated by red arrows. Representative images are shown. Scale bar indicates 100 µm. (**D**) Quantification of the percentage of quiescent surfaces per BS. *Dmp1-Cre*$^{-}$;*Tgif1*$^{fl/fl}$, veh n=8; *Dmp1-Cre*$^{-}$;*Tgif1*$^{fl/fl}$, PTH n=8, *Dmp1-Cre*$^{+}$;*Tgif1*$^{fl/fl}$, veh n=8; *Dmp1-Cre*$^{+}$;*Tgif1*$^{fl/fl}$, PTH n=8. Data are presented as mean ± SEM. One-way ANOVA with Tukey's multiple comparison test, *p<0.05, **p<0.01 vs Tgif1$^{+/+}$, or veh (B) or *Dmp1-Cre*$^{-}$;*Tgif1*$^{fl/fl}$, veh (D) .

The online version of this article includes the following source data for figure 6:

**Source data 1.** Numerical data related to *Figure 6B and D*.

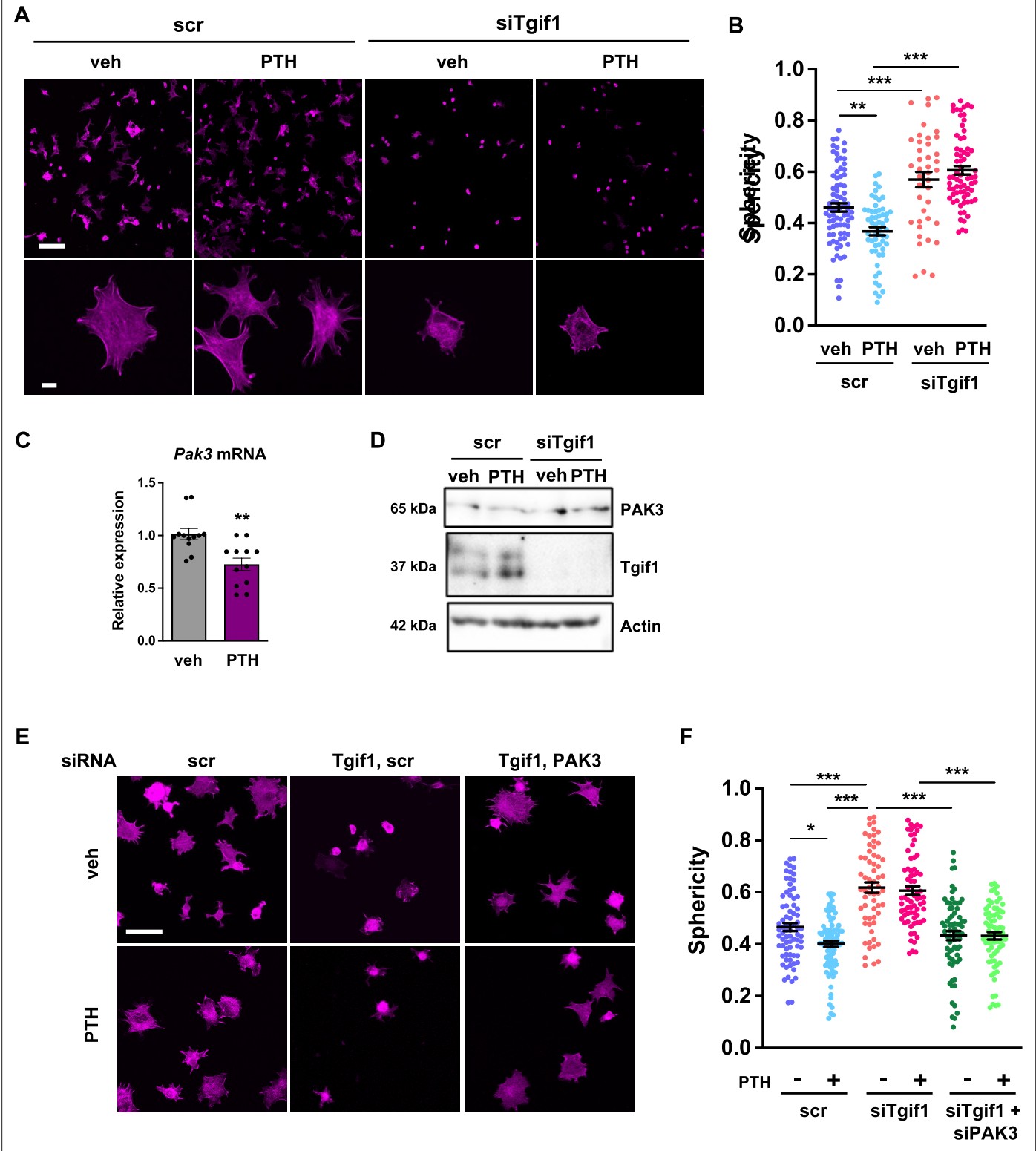

**Figure 7.** PTH promotes cell spreading via TG-interacting factor 1 (Tgif1)-PAK3 signaling. (**A**) OCY454 cells were transfected with siRNA against *Tgif1* (siTgif1) or scramble control siRNA (scr) and treated with vehicle (veh) or PTH for 4 hr. Cells were allowed to adhere on Col-I coated slides for 60 min. The cytoskeleton was visualized by phalloidin staining (magenta). Scale bars indicate 100 μm (upper panel) or 10 μm (lower panel). (**B**) Cell sphericity was quantified using IMARIS software. One-way ANOVA with Tukey's multiple comparisons test was applied, **p<0.01, ***p<0.001. (**C**) p21-activated kinase 3 (*Pak3*) mRNA expression after 4 hr of PTH treatment. Unpaired t-test, **p<0.01 vs. veh. (**D**) Representative image of immunoblots demonstrating

*Figure 7 continued on next page*

*Figure 7 continued*

PAK3 and Tgif1 protein abundance upon PTH treatment in cells transfected with siTgif1 or scr. Actin was used as a loading control. n = 4 independent experiments. (**E**) OCY454 cells were transfected with scr or *Tgif1* siRNA for 48 hr and *Pak3* or scr siRNA for 24 hr and treated with vehicle (veh) or PTH for 4 hr. Cells adhered on Col-I coated slides for 60 min. The cytoskeleton was visualized by phalloidin staining (magenta). Scale bar indicates 50 μm. (**F**) Cell sphericity was quantified using IMARIS software. n=4 independent experiments, one-way ANOVA, Tukey's multiple comparison test, *p<0.05, ***p<0.001. Data are presented as mean ± SEM.

The online version of this article includes the following source data for figure 7:

**Source data 1.** Numerical data related to *Figure 7B, C and F*.

**Source data 2.** Original files for the western blot analysis in *Figure 7D* (anti-PAK3, anti-Tgif1, and anti-actin).

**Source data 3.** PDF containing *Figure 7D* and original scans of the relevant western blot analysis (anti-Tgif1, anti-actin, anti-pERK1/2, anti-ERK1/2, and anti-tubulin) with highlighted bands and sample labels.

Albeit mechanistically distinct, bone repair and PTH treatment have in common that osteoblasts are an important component for the successful execution of bone regeneration and the PTH pharmacological effects. In particular, the guided spatial relocation of activated osteoblasts is a critical feature to exert the respective functions at the right sites in a three-dimensional context since the formation of new bone, regardless of whether in the cortical- or trabecular compartment, occurs in a timely but also in a spatially controlled manner to yield structurally supportive newly formed bone tissue. Various types of cells and tissues of the local microenvironment produce a plethora of signaling factors, which are induced by fractures or PTH treatment. However, little is known about signaling pathways in osteoblasts that receive and integrate these diverse external cues to facilitate the adherence, spreading, and movement of osteoblasts in a coordinated manner. In this context, this study identified the functional importance of PAK3 in Tgif1-deficient osteoblasts. PAK3 expression was increased in the absence of Tgif1, leading to impaired focal adhesion formation and reduced osteoblast spreading. Silencing *Pak3* expression restored osteoblast spreading in Tgif1-deficient cells, demonstrating that PAK3 plays a critical role in mediating the effects of Tgif1 on osteoblast morphology. This work, therefore, contributes to the better understanding of osteoblast dynamics by identifying Tgif1 and PAK3 as downstream signaling regulators that are important for osteoblasts to fulfill motility. Furthermore, the work underscores the physiological relevance of spatial osteoblast dynamics as an integral component of bone remodeling and maintenance of skeletal integrity.

Tgif1 recently emerged as a key determinant of osteoblast activity, bone mass maintenance and an essential component of the bone anabolic PTH signaling pathway (*Haider et al., 2020*; *Saito et al., 2019*) This study also focused on investigating the role of PTH in regulating osteoblast spreading and revealed that PTH treatment facilitated osteoblast spreading via Tgif1-PAK3 signaling. PTH treatment reduced PAK3 expression and increased Tgif1 expression, leading to enhanced osteoblast spreading. Silencing *Tgif1* expression impaired PTH-induced osteoblast spreading, which was restored by simultaneously silencing *Pak3* expression. These new findings on PTH-Tgif1-PAK3 signaling provide novel insights into the function of PTH treatment of bone remodeling and the implication of Tgif1 in this process. As part of being required for bone mass accrual in response to PTH treatment (*Saito et al., 2019*), Tgif1 also contributes to the PTH-induced osteoblast recruitment and subsequent movement to sites where new bone needs to be formed. This adds a spatial component to a functional feature and demonstrates the complexity of bone remodeling and PTH anabolic actions.

This work has the limitation that the findings on the role of Tgif1 in osteoblast spreading and migration were obtained by in vitro assays, which do not necessarily fully reflect in vivo situations. Furthermore, limited evidence exists that the role of Tgif1-PAK3 signaling uncovered in vitro also plays a crucial role in vivo. Thus, future in vivo studies need to further validate our findings for instance by determining *Pak3* expression in *Tgif1*$^{-/-}$ mice in fractures and upon PTH treatment as well as by pharmacologically suppressing *Pak3* expression in Tgif1-deficient mice.

In summary, these findings highlight the pivotal role of Tgif1 in regulating osteoblast morphology, adherence, and migration. Notwithstanding the possibility that other mechanisms exist, our data reveal the involvement of PAK3 in mediating the effects of Tgif1 on osteoblast spreading. The dysregulation of these processes due to Tgif1 deficiency has profound implications for bone remodeling, bone regeneration, and the pharmacological effects of PTH. Reaching a better understanding of the intricate interplay between PTH, Tgif1 and PAK3, provides valuable insights into the molecular

mechanisms governing osteoblast function and could have translational relevance in bone-related disorders and therapies targeting bone health.

# Materials and methods

## Animal models

Mice with germ-line deletion of *Tgif1* or loxP-flanked *Tgif1* loci have been reported previously (*Shen and Walsh, 2005*). To delete Tgif1 in mature osteoblasts and osteocytes, mice expressing the Cre recombinase under the control of the 8 kb fragment of the murine *Dentin matrix protein 1* (*Dmp1-Cre$^{Tg}$*) (*Bivi et al., 2012*) promoter were crossed with mice in which exons 2 and 3 of the Tgif1 gene were flanked by loxP sites (*Tgif1$^{fl/+}$*). Since no bone phenotype was observed in *Dmp1-Cre$^+$; Tgif1$^{+/+}$* mice, *Dmp1-Cre$^-$; Tgif1$^{fl/fl}$* mice were used as controls (*Saito et al., 2019*). In anabolic studies, parathyroid hormone (PTH 1–34; 100 µg/kg of body weight, Biochem) was administered in 8-week-old male mice intraperitoneally five times a week for 3 weeks. To induce conditions of bone healing, 10-week-old male mice of the genotypes *Tgif1$^{+/+}$* or *Tgif1$^{-/-}$* were subject to an open fracture of the tibial midshaft, which was stabilized by an intramedullary pin (*Schindeler et al., 2008*). Three weeks later, mice were sacrificed with subsequent histological examination of the bone tissue. The study received approval by the local authority for animal welfare (Freie und Hansestadt Hamburg, Behörde für Gesundheit und Verbraucherschutz, Nr. 128/13 and 105/15) and experiments were conducted in compliance with all relevant ethical regulations for animal testing and research. For reporting, ARRIVE Guidelines were followed.

## Histology and histomorphometry

Mouse tibiae were collected and fixed in 3.7% PBS-buffered formaldehyde. For histomorphometric analysis, tibiae were embedded in methylmethacrylate. Toluidine blue staining was performed on 4 µm sagittal sections. Quantitative bone histomorphometric measurements were performed on von Kossa and Toluidine blue-stained sections using the OsteoMeasure system (OsteoMetrics). For the quantification of quiescent surfaces (Qs) only surfaces covered by flat cells of intense blue staining without osteoid deposition were considered.

## Micro-computed tomography

Three-dimensional (3D) visualization of the callus formed at the fracture site was performed using micro-computed tomography (µCT). By day 21 upon open fracture of the tibial midshaft, mice were sacrificed and whole legs were harvested followed by fixation in 3, 7% paraformaldehyde. Intramedullary pins were removed prior to imaging to avoid artifacts. Fractured tibiae were scanned using high-resolution µCT with a fixed isotropic voxel size of 10 µm (70 kV at 114 µA, 400 ms integration time; Viva80 micro-CT; Scanco Medical AG). To visualize bone tissue, the threshold value was determined at 326 mg/cm$^3$ hydroxyapatite based on Hounsfield units and a phantom with a linear hydroxyapatite gradient (79–729 mg/cm$^3$). To determine the abundance of mineralized bone within the middle of the callus area, a region of interest was defined and magnified for better visualization.

## Isolation of primary osteoblasts and cell culture

To obtain calvarial osteoblasts, mouse calvariae were dissected from 1 to 3 days old neonatal mice and digested sequentially five times for 25 min in α-MEM (Gibco) containing 0.1% collagenase (Roche) and 0.2% dispase (Roche). Cells obtained from fractions 2–5 were combined according to the genotype and expanded in α-MEM containing 10% FBS (Gibco), 100 U/ml penicillin, and 100 µg/ml streptomycin (P/S, Thermo Fisher Scientific). Long bone osteoblasts were isolated from femora and tibiae of 8-week-old mice. After removing muscles in sterile PBS, bone marrow was flushed, and bones were cut into small pieces. Bone pieces were digested with 0.1% collagenase for 2 hr at 37°C and plated in α-MEM containing 10% FBS and P/S.

## Cell lines

The OCY454 cell line (RRID:CVCL_UW31) was kindly provided by Dr. Paola Divieti Pajevic (Department of Molecular and Cell Biology, Boston University, Boston, USA). Cells were cultured in α-MEM with 10% FBS, 100 U/ml penicillin, and 100 µg/ml streptomycin at 33 °C to permit proliferation and

maintain osteoblast phenotype and at 37 °C for experiments. Cell phenotype was verified by analysis of osteoblast and osteocyte-specific markers. Cells were used at low passages to maintain the phenotype and the absence of mycoplasma was confirmed by regularly performed PCR test.

### 3D cell culture

Long bone osteoblasts were isolated as described above. Bone pieces were digested with 0.1% collagenase for 2 hr at 37°C and plated in α-MEM containing 10% FBS and P/S. Once confluent, outgrowing osteoblasts were seeded on Cellmatrix Type 1 A (Nitta Geratin) placed onto a Millicell membrane (Sigma-Aldrich) in α-MEM with 10% FBS, 100 U/ml penicillin, and 100 µg/ml streptomycin, as reported previously (*Uchihashi et al., 2013*). After 24 hr, the membrane/gel/cells complexes were fixed in 1,5% formaldehyde, embedded in OCT, and frozen. 10 µm sections were cut and stained with Hematoxylin/Eosin and for immunocytochemistry with phalloidin and DAPI as described below. Cell area and perimeter were analyzed from phalloidin-stained sections using the Fiji software.

### Cell spreading and migration assays

For adhesion and spreading assays, cells were incubated in α-MEM with 1% FBS for 4 hr and removed from the culture flasks by gentle trypsinization (Trypsin-EDTA 0.05%, Thermo Fisher Scientific) (*Dejaeger et al., 2017*). Cells were left to adhere for 20, 40, 60, 120, or 240 min on Collagen I-coated eight well-slides ($5×10^3$ cells/well) or plates ($5×10^5$ cells/well). For qPCR, control cells were left in suspension. All spreading assays were performed in serum-free conditions at 37 °C in duplicates or triplicates, which were combined prior to RNA or protein analysis. For the quantification of adhered cells, cells were fixed and stained with hematoxylin or incubated with Calcein–AM 2 µM (Thermo Fisher Scientific) for 60 min. For migration assays, cells were plated at a density of 7500 cellscm$^2$ on Collagen I-coated six-well plates in culture medium (*Dang and Gautreau, 2018*). Cells adhered for 2–3 hr before the medium was changed to α-MEM containing 0.1% FBS. Live cell migration assay was captured using the Improvision LiveCell Spinning Disk microscope. For live cell imaging, five regions of interest/well were selected at time point 0 and images were captured every 10 min for 16 hr. Analysis of migration track length, velocity, and meandering index was performed using the software Volocity 6.

### Cell treatment

For the inhibition of gene transcription, cells were incubated for 15 min with 5 µM of Actinomycin-D (Sigma). For the inhibition of ERK1/2 signaling, cells were incubated for 20 min with 1 µg/ml SCH772984 (Selleckchem) (*Morris et al., 2013*). For the in vitro PTH treatment, cells were incubated with 100 nM PTH (1–34, Biochem) in α-MEM containing 1% FBS for 4 hr prior to the spreading assays.

### RNA isolation and gene expression analysis

Total RNA was isolated using the RNeasy plus kit (Invitrogen) according to the manufacturer's instruction. cDNA was synthesized from total RNA using the ProtoScript First Strand cDNA Synthesis Kit (NEBioLabs). Quantitative real-time PCR was performed using SYBR Green (BioRad). The fold induction of each target gene was calculated using the comparative CT method ($2^{-\Delta\Delta Ct}$). The $C_t$ of each gene of interest was first normalized to the housekeeping gene GAPDH ($\Delta C_t$), and then to the control of the experiment (vehicle or DMSO treatment) to calculate the $\Delta\Delta C_t$. The oligonucleotide sequences used were the following:

| mTgif1 | For GCAGACACACCTGTCCACACTA |
|---|---|
| | Rev GGAATGAAATGGGCTCTCTTCT |
| mPak3 | For CTGAGCAATGGGCACGACTA |
| | Rev AGCCAAAGGAGGTTCCGAAG |
| mPak1 | For TTAGCCGAATCCAGCCTGTCA |
| | Rev TGGTGTTTCTCATCGGAGGG |

*Continued on next page*

*Continued*

| mPak2 | For GCGGTTGTTCCGCTTCC |
|---|---|
|  | Rev AATTATGAAACAGCCAGAGAGGA |
| mPak4 | For GCGCCAAGCCGATGAGTAA |
|  | Rev CTGAGATCTCCACCCGCTTC |
| mCdc42ep1 | For ACGACACGAGGTCTCCACTC |
|  | Rev CCGGGCATCAGCTTTGATTG |
| mCdc42ep2 | For CGCTCCTCAAGCTTCTCAACTC |
|  | Rev AAGGCCAACAAAGACAGGGT |
| mCdc42ep4 | For GAGGCCTGGCCCTCTGATT |
|  | Rev GCACTGATCATCTCGGCTGT |
| mCdc42se2 | For GCTTGGTGTGTGGAGATCCT |
|  | Rev TCAACACTGGCCCTATCGTC |
| mGapdh | For TGCACCACCAACTGCTTAG |
|  | Rev GGATGCAGGGATGATGTTC |

## Immunocytochemistry

Upon adherence of calvarial osteoblasts or OCY454 cells on Collagen I (Roche)-coated surfaces of 8-well chambers, cells were fixed in 1.5% formaldehyde and immune-stained with rabbit anti-Paxillin antibody (1:150, Abcam) or mouse anti-Talin antibody (1:100, Sigma), followed by an incubation with a 488-Alexa Fluor conjugated secondary antibody (1:500, Thermo Fisher Scientific). Hoechst 33258 (Thermo Fisher Scientific) was used at a 1:10,000 dilution for nuclear staining. 633-Alexa Fluor conjugated phalloidin (Thermo Fisher Scientific) was used at a 1:20 dilution to stain actin. Cover slips were mounted with Fluoromount G (Southern Biotech).

## Image acquisition and quantification

Images were acquired using the Zeiss Apotome or the confocal Leica SP5 and SP8 microscopes with 20 X or 63 X objectives. Images for the analysis of focal adhesions were quantified with the Image J software as described previously (*Horzum et al., 2014*). Images for 3D quantification were elaborated by IMARIS software (BITPLANE, Oxford). The Z-stacks obtained by confocal microscopy were rebuilt by IMARIS and the parameters of interest were quantified after removal of the background noise. The value of the sphericity for each cell was provided automatically by the software after cell selection. Quantification of images was performed using raw data. Representative images shown have been optimized equally and only for brightness and contrast.

## Immunoblot analysis

Samples were lysed in low salt RIPA buffer pH 7.5 (50 mM Tris base, 150 mM NaCl, 0.5% Nonidet P-40, 0.25% Sodium deoxycholate,) supplemented with complete protease and phosphatase inhibitors (Roche). Lysates were denatured in Laemmli Sample Buffer (BioRad), resolved by SDS-PAGE, and transferred onto nitrocellulose membranes (GE Healthcare). Membranes were incubated with the following primary antibodies: rabbit anti-Tgif1 antibody (Abcam, 1:500), rabbit anti-PAK3 antibody (Cell Signaling, 1:300), rabbit anti-phospho ERK1/2 antibody (Cell Signaling, 1:1000), mouse anti-ERK1/2 antibody (Cell Signaling, 1:1000), mouse anti-tubulin antibody (Sigma, 1:2000), mouse anti-actin antibody (Millipore, 1:5000), rabbit anti-integrin β1 (Cell Signaling, 1:500), rabbit-anti FAK (Cell Signaling, 1:500), rabbit-anti p-FAK (Cell Signaling, 1:500), rabbit-anti paxillin (Cell Signaling, 1:500), rabbit-anti p-S83 paxillin (Cell Signaling, 1:500), mouse anti-src (Cell Signaling, 1:500), rabbit anti-phospho src (Cell Signaling, 1:500), rabbit anti-p38 (Cell Signaling, 1:500), rabbit anti-phospho p38 (Cell Signaling, 1:500), rabbit-anti LRG5 (Cell Signaling, 1:500) (see table below). Membranes were washed with TBST and incubated either with anti-mouse or anti-rabbit IgG horseradish

peroxidase-conjugated secondary antibody (Promega, 1:10,000). Chemiluminescence images were captured by the BioRad ChemiDoc detection system.

| Antibody name | Provider | Catalogue Number | RRID |
|---|---|---|---|
| Paxillin | Abcam | ab32084 | AB_779033 |
| PAK3 | Cell Signaling | 2609 | AB_2225298 |
| Actin | Millipore | MAB1501 | AB_2223041 |
| TGIF1 | Abcam | ab52955 | AB_882933 |
| phospho p44/42 | Cell Signaling | 4370 | AB_2315112 |
| p44/42 MAPK | Cell Signaling | 9107 | AB_10695739 |
| alpha-Tubulin | Sigma Aldrich | T6199 | AB_477583 |
| pFAK (Tyr397) | Cell Signaling | 3283 | AB_2173659 |
| FAK | Cell Signalling | 3285 | AB_2269034 |
| p38 MAPK | Cell Signaling | 8690 | AB_10999090 |
| phospho p38 | Cell Signaling | 4511 | AB_2139682 |
| p-paxillin | ECM Biosciences | PP1341 | AB_2253333 |
| phospho-Src | Cell Signaling | 6943 | AB_10013641 |
| Anti-Src | Millipore | 05–184 | AB_2302631 |
| Integrin b1 | Cell Signaling | 4706 | AB_823544 |
| LGR5 | LSBio | LS-C205227 | |
| Talin | Sigma-Aldrich | T3287 | AB_477572 |

## Gene silencing

OCY454 cells ($10^6$ cells/reaction) were transfected with the small interfering RNAs (siRNAs) ON-TARGET plus mouse *Tgif1* SMART pool (Dharmacon) or ON-TARGET plus non-targeting control pool (scramble) using NEON electroporation (Invitrogen). ON-TARGET plus mouse *Pak3* SMART pool siRNAs or scrambled were transfected 24 hr later to the same cells using Lipofectamine 3000 (Life Technologies). The final siRNA concentration was 70 nM/sample.

## DNA constructs and luciferase assays

The wild-type human *TGIF1* promoter luciferase reporter constructs, and the *TGIF1* promoter luciferase reporter construct in which the AP1 binding site was mutated have been reported previously (*Saito et al., 2019*). OCY454 cells were transfected with the human *TGIF1* promoter luciferase reporter constructs, or the 6X-TRE-luciferase reporter (6X-TRE-luc; Clontech) to analyze AP-1 activity (*Sabatakos et al., 2008*), or the (–2329/149) rat *Pak3* promoter, provided by Dr. Virna D. Leaner (Institute of Infectious Disease and Molecular Medicine, University of Cape Town, Cape Town, South Africa) (*Holderness Parker et al., 2013*), along with the humanized Renilla reporter plasmid (1:10 ratio) using Lipofectamine 3000. Luciferase assays were performed using the Dual Luciferase Reporter Gene Assay System (Promega) according to the instructions provided by the manufacturer. Firefly luciferase activity was normalized to Renilla luciferase activity.

## Promoter analysis and chromatin immunoprecipitation

Analysis of putative Tgif binding sites on the (–2329/149) rat *Pak3* promoter was performed using the online platform ALGGEN-PROMO. Chromatin immunoprecipitation experiments were performed using the A/G MAGNA CHIP kit (Millipore) according to the manufacturer's instruction. Briefly, $10×10^6$ cells of the OCY454 cell line were plated onto 15 cm² dishes and incubated at 37 °C for 5 days. Cells were then fixed with 1% formaldehyde for 10 min, followed by 5 min of quenching using glycin. After chromatin isolation, DNA was sheared in fragments ranging between 100 and 500 bp in length through 15 cycles of high-frequency sonication using a BioraptorPlus. Immunoprecipitations

were performed using 7 µg of rabbit polyclonal anti-Tgif1 antibody (Santa Cruz Biotechnology) or anti-rabbit ChIP grade IgG (Abcam). The amount of pulled-down DNA was quantified by qPCR. ChIP-qPCR data were first normalized to the input of each precipitation ($2^{(Ct\ 100\%\ input-\ Ct\ sample)}$), and then to the relative IgG sample. The gene *Rarα* was used as a positive control (*Zhang et al., 2009*). The oligonucleotides used to amplify the Tgif1 binding site and as negative control were the following:

| | For | CCAGGAAAATCCTGTTATGCTTCC | | For | GAAATTCTGTGTTGGCCGCA |
|---|---|---|---|---|---|
| BS # 1 | Rev | GGGTTAAAACAGTAGCTACATCCC | Neg | Rev | TCAGCACCTACAATTCTGACCA |

## Statistical analysis

All experiments were performed minimum of three times. Data points in graphs indicate independent biological replicates. Statistical analyses were performed using the statistical package Prism v 9.00 (GraphPad Software, San Diego, CA, USA). Statistically significant differences were determined using the unpaired T-test for comparing two groups. ANOVA followed by Bonferroni or Tuckey Multiple Comparison Test was used to compare more than two groups. The repeated measures ANOVA, estimated margins means test was used to compare the differences of two genotypes over time.

## Acknowledgements

We are grateful to A Gasser und S Schroeder as well as AV Failla and B Zobiak of the UKE Microscopy Imaging Facility (Umif) for conceptual and technical support; G Arndt and P Missberger for mouse husbandry. We thank C Walsh for providing Tgif1-deficient mice, SJ Brandt for providing the Tgif1 reporter constructs, P Divieti Pajevic for kindly contributing the Ocy454 cell line, and V.D. Leaner for providing the rat PAK3 promoter. S Bolamperti was supported by a postdoctoral fellowship from the European Calcified Tissue Society (ECTSABBF2015_0019). H Saito was supported by a postdoctoral fellowship from the Japanese Society for the Promotion of Science. E Hesse received funding from the Deutsche Forschungsgemeinschaft (HE 5208/2–1, HE 5208/2–3 and HE 5208/3–1), the AO-Foundation (S-13–73 H) and the European Commission (PCIG10-GA-2011–303722). H Taipaleenmäki received funding from the Deutsche Forschungsgemeinschaft (TA 1154/1–1 and TA 1154/1–2).

## Additional information

### Funding

| Funder | Grant reference number | Author |
|---|---|---|
| European Calcified Tissue Society | ECTSABBF2015_0019 | Simona Bolamperti |
| Japan Society for the Promotion of Science | | Hiroaki Saito |
| Deutsche Forschungsgemeinschaft | HE 5208/2-1 | Eric Hesse |
| Deutsche Forschungsgemeinschaft | HE 5208/2-3 | Eric Hesse |
| Deutsche Forschungsgemeinschaft | HE 5208/3-1 | Eric Hesse |
| AO Foundation | S-13-73H | Eric Hesse |
| European Commission | PCIG10-GA-2011-303722 | Eric Hesse |
| Deutsche Forschungsgemeinschaft | TA 1154/1-1 | Hanna Taipaleenmäki |

| Funder | Grant reference number | Author |
|---|---|---|
| Deutsche Forschungsgemeinschaft | TA 1154/1-2 | Hanna Taipaleenmäki |

The funders had no role in study design, data collection and interpretation, or the decision to submit the work for publication.

## Author contributions

Simona Bolamperti, Formal analysis, Funding acquisition, Investigation, Visualization, Methodology, Writing - original draft; Hiroaki Saito, Formal analysis, Investigation, Methodology; Sarah Heerdmann, Investigation, Methodology; Eric Hesse, Conceptualization, Supervision, Funding acquisition, Project administration, Writing – review and editing; Hanna Taipaleenmäki, Conceptualization, Data curation, Supervision, Funding acquisition, Project administration, Writing – review and editing

## Author ORCIDs

Simona Bolamperti (iD) http://orcid.org/0000-0002-8940-8859
Eric Hesse (iD) http://orcid.org/0000-0002-2778-7208
Hanna Taipaleenmäki (iD) http://orcid.org/0000-0002-8254-9333

## Ethics

The study received approval by the local authority for animal welfare (Freie und Hansestadt Hamburg, Behörde für Gesundheit und Verbraucherschutz, Nr. 128/13 und 105/15) and experiments were conducted in compliance with all relevant ethical regulations for animal testing and research.

Reviewer #1 (Public review): https://doi.org/10.7554/eLife.94265.3.sa1
Reviewer #2 (Public review): https://doi.org/10.7554/eLife.94265.3.sa2
Author response https://doi.org/10.7554/eLife.94265.3.sa3

# Additional files

## Supplementary files
• MDAR checklist

## Data availability

All data generated or analysed during this study are included in the manuscript and supporting files. Source data files have been provided for all figures.

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
