## [Editor Report · eLife assessment]

This **important** work substantially advances our understanding of osteoblast migration to the sites of bone formation and regeneration. The evidence supporting the conclusion is **compelling**, with rigorous in vitro assays for cellular and biochemical aspects and with appropriate in vivo models. The work will be of broad interest to developmental biologists and bone biologists.

---

## [Referee Report · Reviewer #1 (Public review)]

Summary:

The authors were trying to achieve that Tgif1 expression is regulated by EAK1/2 and PTH in a time-dependent manner, and its roles in suppressing Pak3 for facilitating osteoblast adhesion. The authors further tried to achieve that the Tgif1-Pak3 signaling plays a significant role in osteoblast migration to the site of bone repair and bone remodeling.

Strengths:

- In a previous study, they demonstrated that Tgif1 is a target gene of PTH, and the absence of Tgif1 failed to increase bone mass by PTH treatment (Saito et al., Nat Commun., 2019). In this study, they found that Tgif1-Pak3 signaling prompts osteoblast migration through osteoblast adhesion to prompt bone regeneration. This novel finding provides a better understanding of how Tgif1 expression in osteoblasts regulates adherence, spreading, and migration during bone healing and bone remodeling.

- The authors demonstrated that ERK1/2 and PTH regulate Tgif1 expression in a time-dependent manner and its role in suppressing Pak3 through various experimental approaches such as luciferase assay, ChIP assay, and gene silencing. These results contribute to the overall strength of the article.

Weaknesses:

None after substantial revisions especially in vivo parts.

---

## [Referee Report · Reviewer #2 (Public review)]

Summary:

Bolamperti S. et al. 2023 investigates whether expression of TG-interacting factor (Tgif1) is essential for osteoblastic cellular activity regarding morphology, adherence, migration/recruitment, and repair. Towards this end, germ-line Tgif1 deletion (Tgif1-/-) mice or male mice lacking expression of Tgif1 in mature osteoblastic and osteocytic cells (Dmp1-Cre+; Tgif1fl/fl) and corresponding controls were studied in physiological, bone anabolic, and bone fracture-repair conditions. Both Tgif1-/- and Dmp1-Cre+; Tgif1fl/fl exhibited decreased osteoblasts on cancellous bone surfaces and adherent to collagen I-coated plates. Tgif1-/- mice exhibit impaired healing in the tibial midshaft fracture model, as indicated by decreased bone volume (BV/Cal.V), osteoid (OS/BS), and low osteoblasts (number and surface). Likewise, both Tgif1-/- and Dmp1-Cre+; Tgif1fl/fl show impaired PTH 1-34, (100 µg/kg, 5x/wk for 3 wks) osteoblast activation in vivo, as detected by increases in quiescent bone surfaces. Mechanistic in vitro studies then utilized primary osteoblasts isolated from Tgif1-/- mice and siRNA Tgif1 knockdown OCY454 cells to further investigate and identify the downstream Tgif1 target driving these osteoblastic impairments. In vitro, Tgif1-/- osteoblastic and Tgif1 knockdown OCY454 cells exhibit decreased migration, abnormal morphology, and decreased focal adhesions/cell. Unexpectantly though, localization assays revealed Tgif1 to primarily concentrate in the nucleus and not to co-localize with focal adhesions (paxillin, talin). Also, expression of major focal adhesion components (paxillin, talin, FAK, Src etc.) or the Cdc42 family was not altered by loss of Tgif1 expression. In contrast, PAK3 expression is markedly upregulated by loss of Tgif1. In silico analysis followed by mechanistic molecular assays involving ChIP, siRNA (Tgif1, PAK3), and transfection (rat PAK3 promoter) techniques show that Tgif1 physically binds to a specific site in the PAK3 promoter region. Further, the knockdown of PAK3 rescues the Tgif1-deficient abnormal morphology in OCY454 cells. This is the first study to identify the novel transcriptional repression of PAK3 by Tgif1 as well as the specific Tgif1 binding site within the PAK3 promoter.

Strengths:

This work has a plethora of strengths. The co-authors achieved their aim in eliciting the role of Tgif1 expression to osteoblastic cellular functions (morphology, spreading/attachment, migration). Further, this work is the first to depict the novel mechanism of Tgif1 transcriptional repression of PAK3 by a through usage of mechanistic molecular assays (In silico analysis, ChIP, siRNA, transfection etc.). The conclusions are well supported and justified by these findings, as the appropriate controls, sample sizes (statistical power), statistics, and assays were fully utilized.

Claims and conclusions justified by data? Yes. absolutely

Weaknesses:

None. All reviewer comments were fully addressed.

---

## [Author Response]

The following is the authors’ response to the original reviews.

We are very grateful to both Reviewers, the Reviewing Editor and the Senior Editor for carefully reviewing our manuscript and for providing useful comments and suggestions that further improved the quality of our work. We appreciate that our work is perceived to substantially advance the understanding of osteoblast migration and that the experiments are found to be rigorous and to provide conclusive evidence. We also look forward to reaching a broad audience in the field. Below we provide a point-by-point response to each suggestion made by the reviewers and explain how we included their recommendations in the revised manuscript.

**Public Reviews**

**Reviewer #1 (Public Review):**
Summary:The authors were trying to achieve that Tgif1 expression is regulated by EAK1/2 and PTH in a timedependent manner, and its roles in suppressing Pak3 for facilitating osteoblast adhesion. The authors further tried to show that the Tgif1- Pak3 signaling plays a significant role in osteoblast migration to the site of bone repair and bone remodeling.Strengths:In a previous study, it was demonstrated that Tgif1 is a target gene of PTH, and the absence of Tgif1 failed to increase bone mass by PTH treatment (Saito et al., Nat Commun., 2019). In this study, the authors found that Tgif1-Pak3 signaling prompts osteoblast migration through osteoblast adhesion to prompt bone regeneration. This novel finding provides a better understanding of how Tgif1 expression in osteoblasts regulates adherence, spreading, and migration during bone healing and bone remodeling.The authors demonstrated that ERK1/2 and PTH regulate Tgif1 expression in a time-dependent manner and its role in suppressing Pak3 through various experimental approaches such as luciferase assay, ChIP assay, and gene silencing. These results contribute to the overall strength of the article.

We thank the reviewer for acknowledging the novelty of our findings as well as the strength of the manuscript.

Weaknesses:The authors need to further justify why they focused on Pak3 in the introduction by mentioning its known function for cell adhesion.

We thank the reviewer for this suggestion. We mention in the introduction that we further investigated Pak3 due to its implication in cell adhesion (page 6, lines 7-8).

Some results indicated statistically significant but small changes. The authors need to explain in the discussion part why they believe this is the major mechanism or why there may be some other possible mechanisms.

We agree with this comment. We are confident that our work identified an important mechanism by which Tgif1 regulates cellular features of osteoblasts. However, it is certainly possible that other mechanisms may exist as well. We discuss this point in the revised manuscript (page 18, lines 16-17).

The study does not include enough in vivo data to claim that this mechanism is crucial for bone healing and bone remodeling in vivo.

Re: We agree with this point and have modified the abstract accordingly by replacing “crucial” with “implicated in” as well as the text by changing “crucial” to “important” (page 2, line 9). Furthermore, we discuss this limitation in the revised manuscript (page 18, lines 9-14).

**Reviewer #2 (Public Review):**
Summary:Bolamperti S. et al. 2023 investigate whether the expression of TG-interacting factor (Tgif1) is essential for osteoblastic cellular activity regarding morphology, adherence, migration/recruitment, and repair. Towards this end, germ-line Tgif1 deletion (Tgif1-/-) mice or male mice lacking expression of Tgif1 in mature osteoblastic and osteocytic cells (Dmp1-Cre+; Tgif1fl/fl) and corresponding controls were studied in physiological, bone anabolic, and bone fracture-repair conditions. Both Tgif1-/- and Dmp1-Cre+; Tgif1fl/fl exhibited decreased osteoblasts on cancellous bone surfaces and adherent to collagen I-coated plates. Tgif1-/- mice exhibit impaired healing in the tibial midshaft fracture model, as indicated by decreased bone volume (BV/Cal.V), osteoid (OS/BS), and low osteoblasts (number and surface). Likewise, both Tgif1-/- and Dmp1-Cre+; Tgif1fl/fl show impaired PTH 1-34, (100µg/kg, 5x/wk for 3 wks) osteoblast activation in vivo, as detected by increases in quiescent bone surfaces. Mechanistic in vitro studies then utilized primary osteoblasts isolated from Tgif1-/- mice and siRNA Tgif1 knockdown OCY454 cells to further investigate and identify the downstream Tgif1 target driving these osteoblastic impairments. In vitro, Tgif1-/- osteoblastic and Tgif1 knockdown OCY454 cells exhibit decreased migration, abnormal morphology, and decreased focal adhesions/cells. Unexpectantly though, localization assays revealed Tgif1 to primarily concentrate in the nucleus and not to co-localize with focal adhesions (paxillin, talin). Also, the expression of major focal adhesion components (paxillin, talin, FAK, Src, etc.) or the Cdc42 family was not altered by loss of Tgif1 expression. In contrast, PAK3 expression is markedly upregulated by loss of Tgif1. In silico analysis followed by mechanistic molecular assays involving ChIP, siRNA (Tgif1, PAK3), and transfection (rat PAK3 promoter) techniques show that Tgif1 physically binds to a specific site in the PAK3 promoter region. Further, the knockdown of PAK3 rescues the Tgif1-deficient abnormal morphology in OCY454 cells. This is the first study to identify the novel transcriptional repression of PAK3 by Tgif1 as well as the specific Tgif1 binding site within the PAK3 promoter.Strengths:This work has a plethora of strengths. The co-authors achieved their aim of eliciting the role of Tgif1 expression in osteoblastic cellular functions (morphology, spreading/attachment, migration).Further, this work is the first to depict the novel mechanism of Tgif1 transcriptional repression of PAK3 by a thorough usage of mechanistic molecular assays (in silico analysis, ChIP, siRNA, transfection etc.). The conclusions are well supported and justified by these findings, as the appropriate controls, sample sizes (statistical power), statistics, and assays were fully utilized.The claims and conclusions are justified by the data.

Re: We are grateful to this reviewer for recognizing the novelty, strengths, and rigor of our study and for acknowledging that the data convincingly support the conclusions drawn.

Weaknesses:The discussion section could be expanded with a few sentences regarding limitations to the current study and potential future directions.

Re: In the revised manuscript, we are discussing limitations of the work and describe possible future directions (page 18, line 9-14).

**Recommendations For The Authors:**

**Reviewer #1 (Recommendations For The Authors):**
(1) The cell spreading and migration assay is quite artificial. Trypsinized osteoblasts and quiescent osteoblasts are totally different. The authors need to cite papers from other groups to justify whether the cell spreading and migration assay is appropriate to achieve the goals of this study.

Re: The reviewer is right that in vitro assays are often artificial and do not necessarily fully reflect in vivo situations. We have taken this aspect into account and discuss it in the revised manuscript (page 18, lines 9-10). In addition, we have included references from other groups who have used similar assays to study cell spreading and migration (Dejaeger M et al., 2017 and Dang et al., 2018).

(2) Page 13 Line 15: The statement "Osteoblasts are greatly impaired in the ability to migrate into the repair zone" is an overstatement. The experiments in Figure 5 do not necessarily reflect osteoblast migration activities. The authors need to rephrase the sentence or need to show observation of earlier time points (e.g., 1 week after fracture) in their bone healing experiments. The number of osteoblasts/surface in Tgif1+/+ and Tgif1-/- mice at different time points during bone healing should be a good indicator for the migration of osteoblasts to the repair site.

Re: We understand the critique that a time course or lineage tracing experiments would provide better evidence for the statement of osteoblast migration into the repair zone. To avoid overinterpretations we have removed the sentence from the revised manuscript.

(3) Page 14, Line 24: Regarding the sentence "The observation that Tgif1 is crucial for osteoblast adherence, spreading, and migration", the authors need to clearly mention this statement is based on the in vitro experiments. The animal studies are not enough to claim that the mechanism is crucial for adherence, spreading, and migration.

Re: We thank the reviewer for pointing out this limitation. We have clarified that the finding that Tgif1 is crucial for osteoblast adherence, spreading and migration was made in vitro (page 14, line 22).

(4) The authors need to demonstrate the suppression of Pak3 expression in PTH-treated mice in vivo, in addition to the in vitro culture system (Fig. 7C and 7D).

Re: We agree with the reviewer that this experiment would be very insightful. However, this is beyond the scope of the current work. Nevertheless, to take this valid point into consideration, we mention it in the discussion as potential future direction (page 18, lines 11-14).

(5) The authors need to demonstrate that the pharmacologic suppression of Pak3 in Tgif1-/- mice reduces the % of quiescent surface/BS in vivo.

Re: This point is also well taken, and we agree that a suppression of Pak3 in Tgif1-deficient mice would be very informative to support our in vitro findings. However, this may also be part of future investigations. This is emphasized in the discussion of the revised manuscript (page 18, lines 11-14).

Figures (Minor)Fig. 1:Fig. 1AArrows need to indicate a more precise position.

Re: The position of the arrows has been optimized.

Fig. 1DEWhat are blue/red bars (genotypes)?

Re: The colors indicate the genotypes. A legend has been added to the revised figure.

Fig. 1KQuantification data is needed.

Re: Thank you for this suggestion. We added a quantification of the data (Fig. 1L, M; page 8, lines 3-4; page 21, lines 5-6)

Fig. 2AShow the representative high-magnification image of round (non-spread) cells.

Re: Representative high-magnification images (insets) are provided in the revised figure 2A.

Fig. 5Red arrows need to indicate a more precise position.

Re: The arrows have been repositioned.

Fig. 6A, CRed arrows need to indicate a more precise position.

Re: The arrows have been repositioned.

**Reviewer #2 (Recommendations For The Authors):**
(1) The microscopy images and analyses are excellent.

Re: We thank the reviewer for acknowledging the quality of our microscopy studies.

(2) Since the Tgif1-/- mouse has low osteoclast numbers, is it possible that this is a contributing factor to the delays/impairment in bone healing, given that resorption also has a role in fracture repair? Since the focus of these studies is on osteoblastic cells, this point is a little out of scope. However, would the authors consider exploring this further in the discussion section?

Re: This point is well taken by the reviewer, and we agree that osteoclasts could certainly play a role in the impaired fracture healing. To acknowledge this aspect, we followed the recommendation and discuss this aspect in the revised manuscript (page 16, lines 22-24).

RevisionsWould the authors consider slightly re-wording the title? Tgif1 suppresses PAK3 expression; however, Tgif1-deficiency leads to the unregulated elevation of PAK3 expression.

Re: Thank you for pointing this out. We agree with the reviewer and adapted the title accordingly.

Suggestions(1) Is it possible that apoptosis and/or anoikis is being induced by Tgif1 deficiency in osteoblastic cells?

Re: We do not have data towards this direction and although Tgif1-deficient osteoblasts are overall viable and well expanding, we cannot fully exclude this possibility.

(2) For the fracture study, any differences in overall callus size? Would it be possible to perform micro-CT imaging with some of these samples?

Re: There is no difference in non-mineralized callus size between Tgif1+/+ and Tgif1-/- mice. However, there is less mineralized bone per callus area in Tgif1-/- mice, confirming an impaired osteoblast phenotype. As suggested by the reviewer, we added representative micro-CT images and the respective information to the revised manuscript (Fig 5F; pages 19-20).

(3) Fracture repair experiment-is PAK3 expression downregulated with fracture injury; and/or, is PAK3 upregulated by loss of Tgif1 expression?

Re: Unfortunately, we do not have data to answer this very interesting question and it would need to be addressed in future studies. This is mentioned in the revised discussion (page 18, lines 12-14).

(4) Fig 7F. within PTH treated cells, is the light blue SCR sphericity statistically different than the light green siTgif1 + siPAK3 ? While the statement of the "lack of both, Tgif1 and PAK3 prevented PTH-induced decrease in cell sphericity" is supported by the lack of differences between dark green vs. light green; is it also possible that this is due to the siPAK3 returning sphericity to control (scr) levels? (i.e. hitting a floor limit of detection).

Re: We thank the reviewer for this thoughtful question. There is no statistically significant difference between light blue and light green. Silencing PAK3 restores the impaired capacity to spread that occurs in the absence of Tgif1 to the level of scr controls (significant difference between dark and light red vs. dark and light green and no difference between either dark or light blue vs. dark or light green). However, unlike in the (scr) controls, in the absence of both Tgif1 and PAK3, the cells do not respond to PTH (statistically significant difference between dark and light blue, no difference between dark and light green). Based on the data, cells can reach sphericity of less than 0.2 and thus it is unlikely that sphericity is “hitting the floor level of detection” in these groups.